# `StarEmbed`: Benchmarking Time Series Foundation Models on Astronomical Observations of Variable Stars

Weijian Li [* 1 2 3]  Hong-Yu Chen [* 1 2 3]  Nabeel Rehemtulla [* 2 4 5]  Ved G. Shah [2 4 5]  Dongho Kim [6]  Dennis Wu [1 2 3]
Qinjie Lin [1 2 3]  Adam A. Miller [2 4 5]  Han Liu [1 2 3 6]

## Abstract

Current time series foundation model (TSFM) training corpora largely omit data with certain complexities like irregular temporal sampling. Astronomical time series of stellar fluxes ("light curves") are available in immense quantities and exhibit irregular sampling, multiple variates, and heteroskedasticity. We introduce `StarEmbed`, the first public benchmark for light curves comprised of real observations of $\sim$40,000 stars across seven classes and evaluations in clustering, classification, and out-of-distribution (OOD) source detection. We benchmark TSFMs with differing architecture and training strategies as well as domain-specific transformers. Our results demonstrate that the `Chronos` family, despite being pre-trained on regularly sampled non-astronomical data, yields state-of-the-art (SOTA) performance in light curve clustering and OOD detection. While no TSFM strictly surpasses the classification performance of the long-established domain baseline, they do demonstrate excellent generalization abilities. `StarEmbed` marks a step toward universal light curve embeddings and improved TSFM performance on challenging data.

## 1. Introduction

The adoption of time series foundation models (TSFMs), with pre-training corpora that span commerce, finance, electricity, and traffic data, is proliferating due to their highly capable, general-purpose representation learning capabilities (Zhou et al., 2021; Nie et al., 2023; Yang et al., 2024; Woo et al., 2024). The pre-training corpora, however, do not include the large quantities of publicly available astronomical time series which contain signatures that are rare in standard benchmarks: multiple variates, irregular temporal sampling, regular and irregular gaps, and heteroscedasticity (see Figure 1). These challenges are unavoidable for the ground-based survey observatories which are generating petabyte-scale time series data sets, such as the Zwicky Transient Facility (ZTF; Bellm et al., 2019) and the Vera C. Rubin Observatory (Rubin; Ivezić et al., 2019). Over 7 yr of operations, ZTF alone has produced multi-variate time series for $\sim$10^9 stars, each containing $\sim$10^3 observations; during its first decade of full survey operations starting in 2026, Rubin will scale this by a factor of $10 - 100\times$. The different variates in a light curve represent different colors of starlight captured by placing a filter (or "passband" / "band") in the telescope's focal path, and the bands' relative values (as a function of time) hold nuanced astrophysical insights into the star. Challenges similar to those seen in light curves are also present in time series originating from other scientific domains, so there is both a pressing need and a unique opportunity to evaluate TSFMs on these real scientific data.

We focus this benchmark on *periodic variable stars*: stars which exhibit brightness variations over regular, periodic intervals. These stars are uniquely valuable probes of astrophysics from stellar interiors to galactic structure and more (e.g., Feast & Walker, 1987; Clementini et al., 2003; Genovali et al., 2014; Catelan & Smith, 2015; Ripepi et al., 2017). Despite the abundance of light curves, there is no standardized benchmark for assessing their embeddings. The absence of common datasets, class sets, and train-test splits has hindered fair, reproducible comparisons and obscured whether domain-specific pipelines outperform generic representations from foundation models (cf., Pan et al., 2024).

---

*Equal contribution  [1]Center for Foundation Models and Generative AI (CFMG), Northwestern University, Evanston, USA [2]NSF – Simons AI Institute for the Sky (SkAI), Chicago, USA [3]Department of Computer Science, Northwestern University, Evanston, USA [4]Center for Interdisciplinary Exploration and Research in Astrophysics (CIERA), Northwestern University, Evanston, USA [5]Department of Physics and Astronomy, Northwestern University, Evanston, USA [6]Department of Statistics and Data Science, Northwestern University, Evanston, USA. Correspondence to: Weijian Li <weijianli@u.northwestern.edu>, Hong-Yu Chen <charlie.chen@u.northwestern.edu>, Nabeel Rehemtulla <nabeelr@u.northwestern.edu>, Adam A. Miller <amiller@northwestern.edu>, Han Liu <hanliu@northwestern.edu>.

*Proceedings of the 43^{rd} International Conference on Machine Learning*, Seoul, South Korea. PMLR 306, 2026. Copyright 2026 by the author(s).

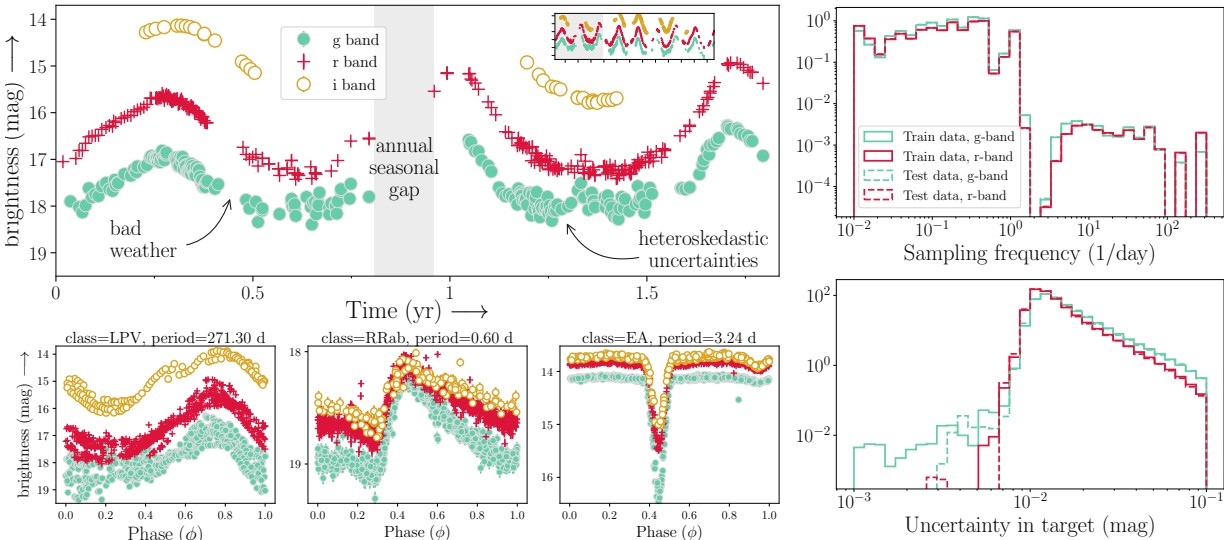

*Figure 1.* Data challenges represented in our data which are characteristic of astronomical time series. *Upper left panel*: Observed light curve of a periodic variable star exhibiting multiple variates, heteroskedastic uncertainties, and large regular and irregular gaps. The inset shows the full ~6.5 years of observations by ZTF. *Lower left panels*: Phase-folded light curves highlighting the periodic patterns in three different classes. Note that most stars have few $i$-band observations (orange points) so we exclude these data from our analysis. *Upper right panel*: Histogram of sampling intervals (days between consecutive measurements). *Lower right panel*: Histogram of measurement uncertainties. Sampling frequency and uncertainty values vary across multiple orders of magnitude, reflecting the considerable influence of these characteristics on the signals in the data.

We introduce `StarEmbed`, the first public benchmark for rigorous, standardized evaluation of SOTA TSFMs on astronomical observations. `StarEmbed` integrates multi-band ZTF light curves for ~40k stars with expert-vetted labels over seven astrophysical classes. Embeddings are evaluated on unsupervised clustering, supervised classification, and OOD source detection in zero-shot and fine-tuned settings. Here, we present results for three TSFM families (`Moirai`, `Chronos`, and `Time-MoE`), a domain-specific transformer (`Astromer`), and the long-standing domain baseline of hand-crafted feature extraction.

We describe related works and models (Section 2), our dataset (Section 3), methodologies (Section 4), results (Section 5), and interpretations of our results (Section 6).

## 2. Related Works and Models

The classification of periodic variable stars has long been a task of significant interest. As such, both pre- (e.g., Debosscher et al., 2007) and post-deep learning models (e.g., Moreno-Cartagena et al., 2025) have been thoroughly explored. Here, we briefly summarize the evolution of the models used in the astrophysics literature, the TSFMs we benchmark, and the baselines we compare them against.

### 2.1. Supervised Classifiers

The first machine learning models to classify variable stars used manually engineered features combined with support vector machines (Debosscher et al., 2007) or gradient boosted decision trees (Richards et al., 2011; Sesar et al., 2017; Boone, 2019). The set of extracted features varies

across studies but is consistently dozens large and includes Fourier coefficients, variability amplitude and skewness, and goodness-of-fit statistics for matches to model templates (Nun et al., 2015; Kim & Bailer-Jones, 2016; Malanchev et al., 2021). Deep learning methods have also been used to eliminate feature engineering by adopting recurrent neural networks (RNNs) (Muthukrishna et al., 2019; Becker et al., 2020; Shah et al., 2025) and transformers (Cabrera-Vives et al., 2024; Moreno-Cartagena et al., 2025). Deep learning efforts, however, do not perform meaningfully better than hand-crafted feature extraction: accuracies of RNN models are $\pm 1 - 3\%$ of hand-crafted features across multiple variable star datasets (see, e.g., Naul et al., 2018). As a result, we choose to use hand-crafted features to establish our baseline performance.

**Baseline Model:** We extract features from the light curves with the `FATS` (Nun et al., 2015) and `light_curve` (Malanchev et al., 2021) software packages. Example features include: the best-fit Lomb-Scargle (Lomb, 1976; Scargle, 1982) period, the scatter, skewness, kurtosis, and other variability statistics. In total, we define 69 features per band, yielding a total embedding size of 138 for the two-band ZTF data (see Appendix C for a full feature list with explanations). In Appendix B.4 we present relative feature importance statistics which find that the period of variability contributes the most to the embedding vector's classification capabilities. Before passing the 138-size embedding to task heads, we normalize each feature to have zero mean and unit variance. While very effective, hand-crafted features

rely heavily on domain knowledge, can be brittle to data quality issues, and are expensive to compute.

## 2.2. Astrophysics Embedding Models

Semi-supervised and self-supervised techniques have also recently been applied to light curves to learn general representations of the data and to later perform downstream tasks. Many of these have been limited to single categories of astrophysical phenomena, for instance, variational autoencoders (Villar et al., 2020), sparse autoencoders (Dillmann et al., 2025), and contrastive learning (Zhang et al., 2024) have been applied to explosive transients. A few foundation models have been developed for variable star light curves, such as FALCO (Zuo et al., 2025) and Astromer (Donoso-Oliva et al., 2023; Donoso-Oliva et al., 2026). Unlike FALCO, Astromer-1 and Astromer-2 are designed to generalize across observatories, so we adopt them as domain-specific foundation models for discerning what advantages in-domain pre-training yields.

**Astromer** (Donoso-Oliva et al., 2023; Donoso-Oliva et al., 2026) Astromer-1 was pre-trained using self-supervised learning on 1.5 million single-band light curves from the MACHO survey (Alcock et al., 2000). The model's output is a fixed-length embedding of 256-dimensions taken from the final attention layer. Astromer-2 functions very similarly but increases the number of model parameters from $0.66$M to $5.4$M and adopts an uncertainty-weighted loss function for pre-training.

## 2.3. Time Series Foundation Models

TSFMs have been shown to consistently outperform the traditional *one-dataset-per-model* schema in multiple fields, including finance, climate science, and commerce (e.g., Yue et al., 2022; Woo et al., 2024; Ansari et al., 2024). With strong performance that scales with model and dataset size, they are a promising tool for driving the future of AI for time series (Edwards et al., 2024; Pan et al., 2024). Astronomy, despite having an enormous collection of light curves, has yet to explore the potential of TSFMs, which may prove transformative in the ability to accomplish multiple downstream tasks. Furthermore, these light curve datasets provide a unique opportunity to evaluate TSFMs with minimal risk of data leakage because they are not included in the widely used pre-training corpora. Each of the TSFMs we consider here have multiple versions with varying parameter counts; in the main text we consider the "small" and "tiny" variants of Moirai and Chronos, respectively, and the smallest available Time-MoE variant, Time-MoE-base. Larger variants are evaluated in Appendix A.3, and we detail computational cost in Table 19.

**Moirai** (Woo et al., 2024; Liu et al., 2025) is a pre-trained TSFM designed for universal forecasting across diverse sampling frequencies, dimensionalities, and data distributions.

Moirai-1 uses a multi-patch-size projection scheme, an any-variate attention mechanism that scales to arbitrary numbers of variables, and a flexible mixture-distribution output head. It is trained on LOTSA, an open archive of 27 billion observations spanning nine domains. Moirai-2 simplifies this framework with a decoder-only architecture and quantile forecasting. By removing masked-token objectives, multi-patch inputs, and mixture-distribution outputs, Moirai-2 emphasizes data efficiency and scalability.

**Chronos** (Ansari et al., 2024) is also a pre-trained TSFM trained for universal forecasting. It treats forecasting as a language-modeling problem by quantizing each time-series point into a fixed vocabulary and training off-the-shelf transformer models (T5-style models with 20M to 710M parameters) with an cross-entropy loss. Augmented by TSMixup and Gaussian-process–generated synthetic data, Chronos is pre-trained on a large collection of public datasets and evaluated on 42 benchmarks. The models deliver strong probabilistic forecasts which are ahead of classical and deep-learning baselines on in-domain data and in zero-shot settings. Chronos-bolt further improves the inference speed by changing from autoregressive decoding to generating multiple forecasting steps in one run. It also changes from point-wise quantization into patching mechanism.

**Time-MoE** (Shi et al., 2025) utilizes a sparse mixture-of-experts architecture to achieve efficient and scalable universal forecasting. It dynamically routes inputs to a small collection of specialized experts rather than relying on dense parameter sharing. This enables scaling up to greater parameter counts with more favorable scaling in computational cost. It is also trained on a massive and heterogeneous time series corpora and demonstrates excellent performance across domains and tasks. Its mixture-of-experts architecture offers an alternative philosophy to a TSFM relative to the Moirai and Chronos families.

## 2.4. Random Embeddings As a Sanity Check Baseline

We generate values for a 256-dimensional vector from a $\mathcal{U}[0, 1]$ distribution as a proxy for a light curve embedding to serve as a marker for the performance floor. These vectors carry no information on the data, so any improvement relative to them suggests the alternative models captured some useful information in their embeddings.

## 3. Dataset

The benchmark dataset includes multi-variate time series observations of periodic variable stars. The data are obtained over 6.5 yr of ZTF operations of imaging the entire Northern sky every few nights and are accessed through the 23rd ZTF data release (DR).[1] ZTF observes in three different

---

[1] https://irsa.ipac.caltech.edu/Missions/ztf.html

bands, $g$, $r$, and $i$[2] (see left side of Figure 1) with effective wavelengths of $\sim$4746Å, 6366Å, and 7829Å, respectively (Dekany et al., 2020). The flux is presented in magnitudes (an astronomy-specific unit), while the time is recorded as the modified Julian date.

The right side of Figure 1 highlights the broadness of sampling frequencies and the significant irregularity in the temporal sampling of this data. The sparsity at high frequency sampling reflects the limited scientific utility of very high frequency observing, and the frequencies of two observations per day mean that observing from the ground can only occur at nighttime and it is always daytime 0.5 days after one observation. The sampling frequency is consistent across the $g$ and $r$ bands and across the train and test splits. Many factors can affect the uncertainties of the flux measurements, which is reflected in their large spread.[3] These features are in stark contrast to those in time series datasets commonly used for TSFM pretraining. LOTSA (Woo et al., 2024), e.g., only includes time series without spread in their sampling frequencies, i.e. only regularly sampled time series, and nearly all time series in LOTSA do not include uncertainties on the target or multiple variates. Therefore, these astronomical time series data represents a new, special out-of-domain test for the TSFMs we explore here.

While many catalogs of periodic variable stars exist in the literature, most of these are produced by low-capacity machine learning models. We instead take the Catalina Surveys Periodic Variable Star Catalog (CSPVS; Drake et al., 2014), a human-labeled catalog created by the Catalina Real-Time Transient Survey (Drake et al., 2009). We extract ZTF DR23 light curves for each CSPVS star. Stars that lack a ZTF light curve are omitted; we also remove (i) observations flagged as "bad" in ZTF DR23; (ii) light curves with $< 32$ total observations; (iii) light curves that lack both $g$ and $r$ observations; and (iv) light curves from classes with fewer than 350 total examples. The resulting dataset contains $\sim$40,000 ZTF light curves of expert-labeled periodic variable stars across seven classes. See Appendix D for a detailed astrophysical description of each class.

Nature produces an imbalance between the number of examples in each class, which is further exacerbated by each class having a different detection efficiency. To ensure that each split gets a representative number of examples from each class, we sample each into the train, validation, and test splits independently in a 7:1:2 ratio. Table 1 shows the counts of examples per class in each split. We release our

*Table 1.* Number of periodic variable stars in our data.

| Class | EW | EA | RRab | RRc | RRd | RS CVn | LPV |
|---|---|---|---|---|---|---|---|
| Train | 18998 | 2889 | 1386 | 3233 | 298 | 942 | 255 |
| Val. | 2690 | 410 | 194 | 463 | 42 | 134 | 35 |
| Test | 5387 | 818 | 397 | 926 | 83 | 276 | 70 |
| Total | 27075 | 4117 | 1977 | 4622 | 423 | 1352 | 360 |

train/validation/test splits on HuggingFace.[4]

### 3.1. ZTF Dataset in Context: Relevance to Other Observatories

Variable star science has an expansive scope that extends beyond ZTF and periodic variables; the `StarEmbed` benchmark is designed to allow for the addition of new datasets and metrics in future expansions. This flexibility is crucial to the long-term health of this benchmark as numerous new time-domain facilities, like Rubin, will begin releasing floods of data in the coming years. Each new survey has unique observational capabilities and priorities that will affect the resulting embeddings and downstream task performance. As the largest astronomical time-domain experiment to date, ZTF is an apt choice for building a preparatory benchmark dataset as ZTF data has already been used to explore emerging AI areas like multi-modality, pre-training, and transformers (e.g., Duev & van der Walt, 2021; Carrasco-Davis et al., 2021; Gagliano et al., 2023; Allam et al., 2023; Zhang et al., 2024; Rehemtulla et al., 2024; 2026).

All current and future datasets associated with the `StarEmbed` benchmark are and will be made public with a license for reuse. We host a leader board tracking model performance on each task on our website[5] as well as an interactive data visualization tool called the StarEmbed Explorer where a user can peruse the data comprising the benchmark.[6] No personal or sensitive information is present in these datasets because they consist of astronomical observations.

## 4. Evaluation Methodology

We assess the quality of embeddings for unsupervised clustering, supervised classification, and OOD source detection. Together, these give a comprehensive view of the intrinsic structure captured by the embeddings and their usefulness for downstream tasks. Below we detail the experimental settings including training procedures and metrics used to evaluate the embeddings. We release our code to reproduce our benchmark experiments.[7] To maintain consistency through-

---

[2] Most ZTF sources have very few or no observations in $i$ band and we therefore exclude it from the benchmark.

[3] We observe a slight over-density of $g$ band measurements with very small uncertainties in the train split. Noting the logarithmic scale on the y-axis, this is not a concern as these represent only a very small number of observations.

[4] https://huggingface.co/datasets/StarEmbed/ZTF_40k

[5] https://www.StarEmbed.com/

[6] https://www.StarEmbed.com/StarEmbed-Explorer/

[7] https://github.com/skai-institute/StarEmbed

out the benchmark, we use identical embedding sizes across different models whenever possible (i.e., when there exists such a pre-trained version of the model). `Astromer-1`, `Astromer-2`, `Chronos-Bolt`, `Chronos` and the Random Embeddings all have embedding size of 256. We adopt the smallest available pre-trained `Moirai` model, `Moirai`, which uses an embedding size of 384.

### 4.1. Unsupervised Clustering

Our unsupervised clustering evaluation assesses the alignment of structure in the embedding's feature space with known periodic variable star classes. Specifically, we use the K-means and Ward's method clustering algorithms, both configured to produce $N = 7$ clusters corresponding to the seven astrophysical classes in the dataset. Before executing the clustering algorithms, we normalize all embeddings to the standard normal because the clustering methods compute Euclidean distances which are sensitive to the scale of entries. The resulting cluster memberships are assigned to class predictions using the Hungarian matching algorithm (Kuhn, 1955).

We produce uncertainties on performance metrics by repeating the K-means algorithm with 10 different initializations, choosing the clustering with the lowest within-cluster variance. Ward's clustering is deterministic, so no uncertainties are provided. We compute three literature-standard metrics to comprehensively evaluate the clustering performance (Huang et al., 2020; Monnier et al., 2020; Sun et al., 2024; Li et al., 2024). **Normalized Mutual Information** (NMI) measures the mutual information between the cluster assignments and the true class labels and is normalized to the range [0,1] with higher values indicating better alignment between the clusters and the true classes. **Adjusted Rand Index** (ARI) evaluates pairwise clustering agreements. It considers how often pairs of light curves are in the same cluster versus the same true class. An ARI of unity indicates perfect clustering, ARI$\sim$0 indicates random clustering, and ARI$<$0 indicates clustering worse than random. **Macro-averaged F1 score** (F1) is the average of the harmonic means of precision and recall computed independently for each class. It treats minority and majority classes equally, and is thus sensitive to the class imbalance inherent to our dataset.

This evaluation setting allows us to probe the separable class structure in the embeddings without any supervised training. We also visualize the embedding spaces with the uniform manifold approximation and projection (UMAP) algorithm, yielding an intuitive, qualitative view of clustering performance in Appendix B.2.

### 4.2. Supervised Classification

Complementary to unsupervised clustering, we also evaluation the embeddings in a supervised classification scenario

where the variable star class labels are directly predicted from the embeddings. We comprehensively evaluate the relevant information content of the embeddings by running this evaluation with four different classification heads: $k$-nearest neighbors ($k$-NN), a logistic linear probe, a random forest (RF), and a multi-layer perceptron (MLP). This simulates a scenario where one uses a pre-trained model to produce embeddings used as feature inputs for a classification task but does not fine-tune the embedding model (hence "zero-shot" in terms of the embedding model).

Both $k$-NN and logistic probes are standard in the embedding evaluation literature (Caron et al., 2021; Zhou et al., 2022; Neelakantan et al., 2022), as they are simple methods that directly reflect separability in the embedding space. RF is included, in part, because it is widely used in the astronomical literature for periodic variable star classification (Naul et al., 2018; Sánchez-Sáez et al., 2021; Pimentel et al., 2022) and it achieves SOTA performance across many datasets (Naul et al., 2018). An MLP is used as a modern, higher capacity, non-linear, deep-learning option. Detailed information on the classifiers, the hyperparameter optimization, and the final hyperparameters for each embedding model can be found in Appendix B.5.

We report standard multi-class classification metrics to summarize the performance of each embedding model and classification head pairing. These include accuracy, the macro-averaged F1 (see Sec. 4.1), precision (the false positive rate subtracted from one), and the recall (the true positive rate).

### 4.3. Out-of-Distribution (OOD) Source Detection

Identifying variable stars physically unlike those in labeled training sets is of great astrophysical interest. To test the effectiveness of the embeddings for detecting such OOD sources, we first create embeddings for the ZTF light curves of CSPVS stars with too few examples to be included in the train/validation/test splits ($\beta$-Lyrae, Blazhko, Anomalous Cepheids, Cepheid-II, HADS, LADS, ELL, Hump, PCEB, and EAup; see Sec. 3). We define these as OOD sources; they are represented in the `StarEmbed` dataset as an additional "anom" data split. The embeddings of the OOD sources are mixed with the test set and run through a "multi-class isolation forest" (MCIF; Gupta et al., 2025) where a separate isolation forest is fit to the embeddings of each of the seven inlier classes in the training set. The minimum of the seven isolation forest scores assigned to a given star's embedding is its OOD score. Isolation forest is a popular method for finding astrophysical outliers (Malanchev et al., 2021), and (Gupta et al., 2025) show that following this multi-class prescription yields better macro-averaged performance than a single isolation forest in many settings, including for periodic variables. Here, the performance is benchmarked with the fraction of sources in the top $N^{\text{th}}$ percentile of OOD scores which are genuine OOD sources:

*Table 2.* Results of unsupervised clustering with K-means and Ward. The best results are highlighted in **bold**, and the second-best results are underlined. The `Chronos` models perform very well on this unseen data, placing first or second in all metrics and universally better than `Moirai` and the `Astromer` models. However, hand-crafted features perform the best overall. Model FLOPs are in Table 19.

| Methods | NMI (K-Means) | ARI (K-Means) | F1 (K-Means) | NMI (Ward) | ARI (Ward) | F1 (Ward) |
|---|---|---|---|---|---|---|
| Astromer-1 | 0.0041(0.0001) | 0.0017(0.0011) | 0.1660(0.0014) | 0.0041 | 0.0001 | 0.1652 |
| Astromer-2 | 0.0082(0.0010) | 0.0192(0.0078) | 0.1590(0.0042) | 0.0091 | 0.0310 | 0.1600 |
| Moirai | 0.1749(0.0017) | 0.0981(0.0028) | 0.2831(0.0034) | 0.1476 | 0.0828 | 0.2612 |
| Moirai-2 | 0.2017(0.0019) | 0.0967(0.0060) | 0.3007(0.0095) | 0.1850 | 0.0603 | 0.2655 |
| Chronos | 0.2374(0.0082) | **0.1596(0.0029)** | 0.3110(0.0362) | 0.1890 | 0.1217 | **0.3671** |
| Chronos-Bolt | 0.2120(0.0033) | 0.1306(0.0125) | 0.3128(0.0027) | 0.2273 | 0.1553 | 0.3662 |
| Time-MoE | 0.2069(0.0108) | 0.0913(0.0090) | 0.3101(0.0227) | 0.1946 | 0.0944 | 0.3385 |
| Hand-crafted Features | **0.2700(0.0058)** | 0.1197(0.0092) | **0.3960(0.0271)** | **0.2508** | 0.1319 | 0.3323 |

*Table 3.* Results of benchmark supervised classification. The best results are highlighted in **bold**, and the second-best are underlined. The $k$-NN and logistic classifiers are deterministic, so we report the 1-run performance; We report the mean and standard deviation of runs with 10 seeds for RF and MLP. The hand-crafted features are SOTA with `Chronos` a clear second-best. See Appendix B.6 for computation analysis of each model.

| Classifier | Metric | Astromer-1 | Astromer-2 | Moirai | Moirai-2 | Chronos | Chronos-Bolt | Time-MoE | Random | HC features |
|---|---|---|---|---|---|---|---|---|---|---|
| $k$-NN | Accuracy | 0.644 | 0.823 | 0.809 | 0.815 | 0.857 | 0.807 | 0.825 | 0.648 | **0.881** |
| | Precision | 0.130 | 0.660 | 0.662 | 0.641 | 0.799 | 0.647 | 0.724 | 0.120 | **0.818** |
| | Recall | 0.141 | 0.489 | 0.509 | 0.546 | 0.623 | 0.542 | 0.560 | 0.140 | **0.661** |
| | F1 | 0.122 | 0.537 | 0.554 | 0.561 | 0.672 | 0.570 | 0.595 | 0.120 | **0.712** |
| logistic | Accuracy | 0.073 | 0.648 | 0.705 | 0.709 | 0.750 | 0.709 | 0.740 | 0.094 | **0.838** |
| | Precision | 0.147 | 0.486 | 0.544 | 0.538 | 0.575 | 0.549 | 0.559 | 0.144 | **0.663** |
| | Recall | 0.165 | 0.668 | 0.680 | 0.678 | 0.730 | 0.676 | 0.694 | 0.128 | **0.854** |
| | F1 | 0.072 | 0.521 | 0.579 | 0.577 | 0.617 | 0.580 | 0.597 | 0.076 | **0.714** |
| RF | Accuracy | 0.676 (0.000) | 0.846 (0.000) | 0.823 (0.001) | 0.820 (0.001) | 0.862 (0.000) | 0.826 (0.001) | 0.839 (0.001) | 0.676 (0.000) | **0.920 (0.001)** |
| | Precision | 0.111 (0.043) | 0.799 (0.006) | 0.716 (0.007) | 0.697 (0.010) | 0.750 (0.056) | 0.707 (0.002) | 0.693 (0.004) | 0.097 (0.000) | **0.866 (0.003)** |
| | Recall | 0.143 (0.000) | 0.526 (0.002) | 0.514 (0.002) | 0.520 (0.004) | 0.597 (0.002) | 0.548 (0.001) | 0.561 (0.003) | 0.143 (0.000) | **0.773 (0.004)** |
| | F1 | 0.115 (0.000) | 0.580 (0.002) | 0.557 (0.002) | 0.556 (0.003) | 0.638 (0.002) | 0.582 (0.001) | 0.598 (0.003) | 0.115 (0.000) | **0.804 (0.003)** |
| MLP | Accuracy | 0.446 (0.147) | 0.627 (0.037) | 0.717 (0.031) | 0.735 (0.019) | 0.783 (0.022) | 0.721 (0.022) | 0.758 (0.025) | 0.308 (0.203) | **0.833 (0.022)** |
| | Precision | 0.154 (0.006) | 0.453 (0.019) | 0.546 (0.022) | 0.560 (0.021) | 0.589 (0.025) | 0.553 (0.026) | 0.579 (0.024) | 0.137 (0.006) | **0.672 (0.025)** |
| | Recall | 0.165 (0.003) | 0.627 (0.020) | 0.722 (0.006) | 0.717 (0.006) | 0.758 (0.006) | 0.696 (0.013) | 0.741 (0.010) | 0.145 (0.002) | **0.851 (0.009)** |
| | F1 | 0.138 (0.020) | 0.470 (0.023) | 0.594 (0.019) | 0.603 (0.015) | 0.643 (0.023) | 0.589 (0.015) | 0.627 (0.019) | 0.094 (0.044) | **0.723 (0.027)** |

the OOD purity.

# 5. Benchmark Results

We highlight the benchmark results below for each of the three downstream tasks: unsupervised clustering, supervised classification, and OOD detection. Besides, we also include extensive analyses and TSFM training experiments.

## 5.1. Unsupervised Clustering

In Table 2 we show that TSFMs generally perform well: (i) the first or second best ranked model in each metric comes from the `Chronos` models; (ii) both `Chronos` models always outperform both `Moirai` models; and (iii) The `Chronos` models and hand-crafted features are highly competitive, frequently alternating between the best and second-best performance. We find the domain-specific `Astromer` models generally yield poor performance, notably always worse than the TSFMs which are not trained on light curves. We further analyze the poor performance of `Astromer-1` in Appendix B.3 and find that its embeddings collapse to similar directions. Appendix B.3 further discusses how this is expected for both `Astromer` models based on results from previous studies. Finally, hand-crafted features achieve the highest global separability (top NMI under both K-means and Ward), reflecting coarse class alignment. In contrast, `Chronos` leads on pairwise consistency (best ARI for K-means, and ARI is more sensitive to pairwise correctness), suggesting that its embeddings form small, pure neighborhoods rather than single, class-wide clusters.

## 5.2. Supervised Classification

Table 9 and the left panel of Figure 2 show that (i) the `Chronos` models perform very well compared to the `Moirai` and `Astromer` models; (ii) `Astromer-2` performs better than the `Moirai` models in some metrics; and (iii) unlike in the clustering results, `Chronos` outperforms `Chronos-Bolt` and achieves second-best performance in nearly all metrics. The hand-crafted features remain superior, yielding an F1 score of $0.804 \pm 0.003$ with the RF classifier. We also find that the RF classifier generally performs better than others, although this is somewhat model-dependent. We conduct a feature importance analysis on the hand-crafted features in Appendix B.4 and find that the periods of the variability in either passband are the most important features.

The two confusion matrices in Figure 2 show that (i) both `Chronos` and the hand-crafted features often confuse RRd sources as RRc; (ii) `Chronos` yields better performance on most classes (EA, RRd, RS CVn, and LPV) although loses overall due to the larger margins in the classes where the hand-crafted features perform better (EW, RRab, RRc).

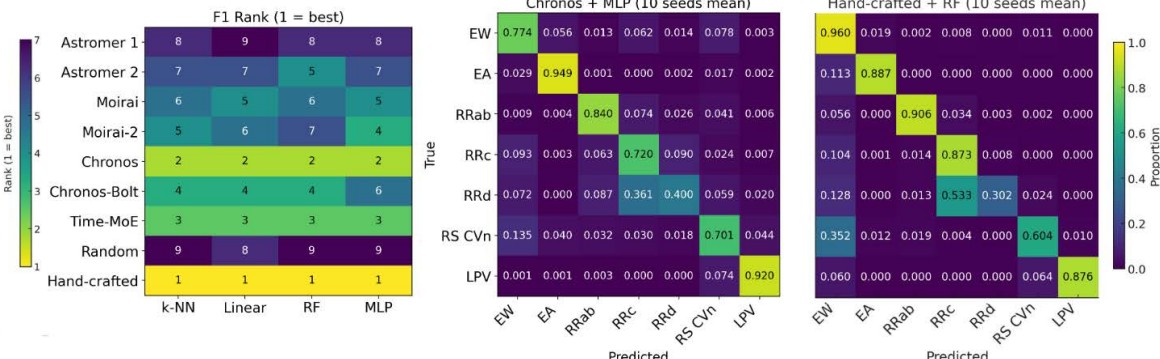

*Figure 2. Left:* F1 Ranking of baselines across classifier heads. Chronos consistently outperforms other TSFMs and the domain-specific Astromer models, the hand-crafted (HC) features perform the best overall. *Right:* Confusion matrix of Chronos + MLP, the best performing TSFM-classifier combination, and of HC features + RF classification, the SOTA baseline in astrophysics. Chronos yields better performance on most classes (EA, RRd, RS CVn, and LPV), indicating TSFM's efficacy in extracting information for classification.

*Table 4.* Purity of top N-percentile of sources selected by out-of-distribution source detection. Best results are highlighted in bold, and the second-best are underlined. Chronos-Bolt ranks first across all metrics with hand-crafted (HC) features being a distant second.

|  | Top 1% | Top 5% | Top 10% |
|---|---|---|---|
| Astromer-1 | 0.017(0.016) | 0.091(0.019) | 0.120(0.001) |
| Astromer-2 | 0.135(0.027) | 0.126(0.006) | 0.120(0.002) |
| Moirai | 0.169(0.024) | 0.143(0.007) | 0.150(0.004) |
| Moirai-2 | 0.143(0.024) | 0.204(0.007) | 0.198(0.004) |
| Chronos | 0.139(0.054) | 0.116(0.028) | 0.149(0.021) |
| Chronos-Bolt | **0.569(0.060)** | **0.532(0.055)** | **0.519(0.038)** |
| Time-MoE | 0.137(0.013) | 0.175(0.006) | 0.176(0.004) |
| Random | 0.116 | 0.116 | 0.116 |
| HC features | 0.213(0.013) | 0.271(0.014) | 0.259(0.003) |

These results show that, despite having never seen astronomical time series, Chronos clearly extracts useful information from the data for supervised classification. The complete set of confusion matrices for all embedding–classifier combinations can be found in Appendix E. We further include recent TSFMs family designing for classification results in Appendix A.1.

### 5.3. Out-of-Distribution Source Detection

Table 4 shows that: (i) Chronos-Bolt is exceptional at isolating OOD sources from the inliers; (ii) the second-overall ranked hand-crafted features deliver considerably worse OOD purity; and (iii) every other model provides only a marginal gain over random selection. Of all the stars evaluated, $\sim 11\%$ are OOD samples. This demonstrates that by applying the MCIF approach to the Chronos-Bolt embeddings, we are able to recover nearly half of the OOD sources by evaluating just 10% of the data, a $\sim 5\times$ improvement in search efficiency over the random embeddings. As these OOD events often correspond to astrophysically rare or anomalous sources, they are of great interest for the astrophysics community. We posit that Chronos-Bolt's patch-based, multi-step generation is less sensitive to step-level variation than Chronos's autoregressive next-token

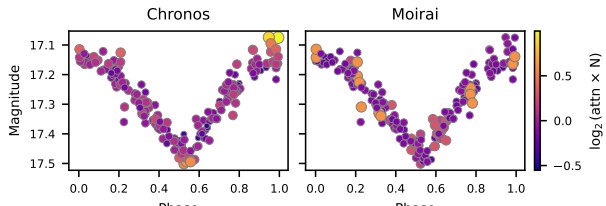

*Figure 3.* Attention score for an RRc light curve, phase-folded with the catalog period. Chronos concentrates attention on the brightness peak, while Moirai's attention is diffuse across the cycle. See detailed setup and more visualizations in Appendix B.1.

training, hence encouraging a tighter inlier manifold and larger off-manifold distances for rare morphologies. This yields weaker clustering of inliers but stronger OOD isolation.

### 5.4. Attention Score Analysis

We further investigate different TSFMs architectures' efficacy in capturing periodic information from the irregular time series. We hypothesize the key to be the tokenization method. Chronos uses point-wise tokenization, whereas Moirai uses patch-wise tokenization. Point-wise tokenization is less destructive when the raw sequence contains irregular gaps and non-uniform local periodic structure, because patching can conflate consecutive observations separated by very different time intervals.

Figure 3 visualizes the attention scores of Chronos and Moirai on one example light curve. Chronos concentrates attention on brightness extrema, thereby capturing the full range of variability in the light curve, including both rapid and slow changes in brightness. In contrast, Moirai groups neighboring measurements into patches. For irregularly sampled light curves, each patch can correspond to dramatically different time baselines, exacerbating the challenge of irregular sampling and making it difficult to learn the underlying variability.

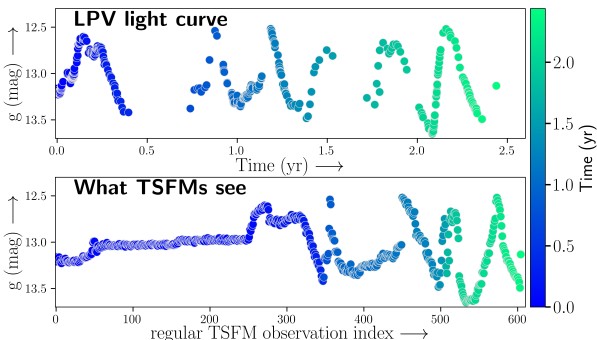

*Figure 4.* A long period variable (LPV) star's light curve (*top*: the real, physical index) and indexed by the order of the measurements (*bottom*: a non-physical index). The coloring corresponds to the physical time index, showing that discarding relative timestamp information causes the TSFMs to receive a warped view of the light curve, limiting their interpretation of the data.

*Table 5.* Classification fine-tuning results for Chronos-tiny on ZTF light curves with an **MLP head**. Same HPO grid and reporting protocol as Table 6. Bold indicates the best in each row.

| Metric | Full FT | LoRA | LayerNorm |
|---|---|---|---|
| Trainable Params | 9.6M (100%) | 1.3M (13.2%) | 1.2M (12.4%) |
| Macro F1 | 0.659 (0.026) | **0.661 (0.015)** | 0.648 (0.016) |
| Accuracy | **0.778 (0.028)** | 0.772 (0.020) | 0.774 (0.023) |
| Macro Prec | **0.613 (0.034)** | **0.613 (0.018)** | 0.605 (0.023) |
| Macro Rec | 0.766 (0.005) | **0.767 (0.006)** | 0.766 (0.006) |

### 5.5. TSFM Failure Analysis

Figure 4 illustrates that the TSFMs evaluated here treat irregularly sampled data as regular, resulting in a warped view of the time series that obscures key physical structures like periodicity (Appendix B.4). That TSFMs perform competitively despite this informational loss suggests that TSFMs with mechanisms specifically architected for irregular sampling and irregular pre-training data will likely unlock significant performance gains. Such a model could drive a paradigm shift from bespoke, fully-supervised pipelines toward off-the-shelf foundational representations for variable star analysis. Pushing analysis frontiers in this way is critical for the era of petabyte-scale light curve data sets. Complimentarily, astronomy has challenging, information-rich time series in game-changing quantities to offer the AI-for-time-series field, and making use of these time series would also help open new applications of TSFMs to other scientific domains with similarly challenging time series.

### 5.6. TSFM Fine-tuning

We also study TSFMs fine-tuning on astrophysical data. Since Chronos is the best-performing TSFM, we fine-tune it on StarEmbed and evaluate it on supervised classification.

**Classification Fine-tuning versus Forecasting Fine-tuning.** We compare two fine-tuning objectives. The first setting, denoted Chronos-FTfcst, uses the orig-

inal Chronos forecasting objective. The second setting, denoted Chronos-FTcls, performs end-to-end classification fine-tuning with an MLP classification head.

Table 11 shows that Chronos-FTfcst does not improve downstream classification and is slightly worse than zero-shot Chronos. In contrast, Chronos-FTcls improves three of the four MLP-based classification metrics over zero-shot Chronos, suggesting that fine-tuning is helpful for Chronos when performed directly on the downstream classification task. The end-to-end classification fine-tuned Chronos still remains below hand-crafted features, however, which indicates that current TSFMs are not yet sufficient to replace even the conceptually simple domain-specific baseline on our challenging scientific dataset.

**Full Fine-tuning versus Parameter-Efficient & Partial Fine-tuning.** Regarding fine-tuning efficiency, we also compare full fine-tuning, LoRA fine-tuning, and partial fine-tuning in which only the LayerNorm parameters are updated. We run each of these experiments with both a linear head and an MLP head.

*Table 6.* Classification fine-tuning results for Chronos on ZTF light curves with a **linear head**. Bold indicates the best in each row.

| Metric | Full FT | LoRA | LayerNorm |
|---|---|---|---|
| Trainable Params | 8.4M (100%) | 102K (1.2%) | 9.2K (0.1%) |
| Macro F1 | **0.661 (0.010)** | 0.622 (0.022) | 0.622 (0.018) |
| Accuracy | **0.777 (0.010)** | 0.746 (0.023) | 0.750 (0.029) |
| Macro Prec | **0.615 (0.012)** | 0.572 (0.026) | 0.571 (0.022) |
| Macro Rec | **0.766 (0.005)** | 0.760 (0.006) | 0.758 (0.004) |

Table 6 shows that with a linear head, full fine-tuning yields the best performance, indicating that when the classification head is linear, updating only a small number of parameters is not sufficient to match full fine-tuning. With an MLP head, full fine-tuning and LoRA yield comparable top performance, despite LoRA tuning fewer than one seventh as many parameters. LayerNorm-only tuning also improves under the MLP head, but remains below LoRA. These results suggest that when only a small number of parameters are updated during fine-tuning, using a more expressive classification head becomes important. Under a linear head, both LoRA and LayerNorm-only tuning appear to be similarly constrained by the limited expressiveness of the head, resulting in lower F1 scores. Under an MLP head, this bottleneck is alleviated by the nonlinear classifier. LoRA also shows a clearer advantage over LayerNorm-only tuning, possibly because LoRA adapts the attention projections and can therefore better focus on discriminative patterns in the time series, whereas LayerNorm-only tuning is limited to scale-and-shift updates. See Appendix A.2 for full results.

*Table 7.* Pre-training TSFMs with a modified RoPE incorporating irregular time gaps and test on sinusoidal synthetic data. We evaluate: (1) RoPE + True timestamp, replacing sequential position indices with actual observation timestamps, (2) vanilla baseline that is the original `Moirai`. Each sample is generated following $x(t) = A\sin(2\pi ft + \varphi) + \varepsilon$. The results indicate the importance of modeling irregular time gaps in the positional encoding. See prediction examples in Figures 5 and 7.

| Metric | RoPE + True timestamp | Vanilla |
|---|---|---|
| $R^2 \uparrow$ | $\mathbf{0.188 \pm 0.102}$ | $0.016 \pm 0.039$ |
| Pearson $\uparrow$ | $\mathbf{0.351 \pm 0.114}$ | $0.202 \pm 0.098$ |
| MSE $\downarrow$ | $\mathbf{0.041 \pm 0.006}$ | $0.050 \pm 0.002$ |
| MAE $\downarrow$ | $\mathbf{0.159 \pm 0.017}$ | $0.183 \pm 0.007$ |

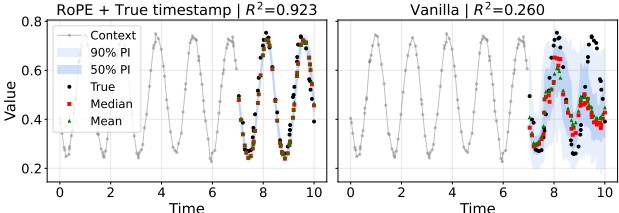

*Figure 5.* Forecasting visualization of Table 7. Encoding the time information correctly into `Moirai` through RoME make perfect prediction on testing data, while vanilla `Moirai`'s predictions has the wrong frequency.

### 5.7. TSFM Architecture Improvement

We also study architectural improvements of TSFMs to identify directions for developing more capable TSFMs natively aligned with irregularly sampled time series. A central limitation observed throughout our experiments is that current TSFMs do not properly treat irregular sampling (see Figure 4). Existing positional encoding are typically applied to regular token or patch indices, but in astronomical light curves, adjacent observations in the input sequence correspond to non-uniform time intervals.

We modify Rotary Positional Encoding (RoPE) in `Moirai` by incorporating the true observation timestamps instead of using the patch index. Table 7 and Figure 5 show that our modification outperforms the vanilla version which does not properly handle the irregular sampling. These observations isolate an important design option for future development and pre-training of TSFMs capable of properly treating irregular temporal sampling. See more experimental details and forecasting examples in Appendix A.5.

## 6. Discussion and Conclusions

We introduced `StarEmbed`, a public benchmark of TSFMs on real multi-band stellar light curves, whose irregular sampling and heteroskedasticity differ substantially from typical TSFM pre-training corpora. By harmonizing expert-vetted labels with ∼40,000 ZTF light curves, we provide a rigorously curated seven-class dataset and conduct comprehensive benchmarking experiments targeting the follow-

ing categories: (i) TSFM zero-shot performance on three key downstream astrophysics tasks (unsupervised clustering, supervised classification, and OOD source detection); (ii) interpretibility analysis (TSFM embedding clusterings and quality analysis, TSFM attention score analysis, failure analysis); (iii) study of TSFM training for irregular time series: TSFM fine-tuning, TSFM architecture design for better pre-training, and TSFM scaling on astrophysics data. The conclusions are as follows.

**TSFM Performance Benchmark (Sections 5.1 to 5.3).** We compare domain-specific embeddings, SOTA general-purpose TSFMs, and the long-standing and top-performing method in astrophysics, hand-crafted features. For clustering, `Chronos` matches hand-crafted features. For supervised classification, hand-crafted features perform best, with `Chronos` consistently second with a small margin. For OOD detection, `Chronos-Bolt` significantly outperforms hand-crafted features. Across all tasks, TSFMs generally outperform the domain-specific transformer, `Astromer`.

**Interpretibility Analysis.** Both TSFMs embedding clusterings and quality analysis show the efficacy of zero-shot TSFMs' embeddings in capturing the differences between different classes of astrophysics data. The failure analysis shows that all TSFMs fall short in capturing periodic information from the irregular time series compared to hand-crafted features. The attention score analysis further unveils that `Chronos`'s model architecture could retain much better periodic information compared to `Moirai`'s.

**Study of TSFM Development on Irregular Time Series.** Beyond benchmarking zero-shot TSFMs, we also study TSFM training, especially on our irregular time series data. Our fine-tuning experiment shows that fine-tuning with a classification head yield a better results on the downstream classification task compared to fine-tuning with a forecasting objective as in pre-training. In terms of pre-training TSFMs from scratch, we demonstrate that a modified RoPE encoding that incorporates the irregular time gaps significantly improves TSFMs ability in capturing periodic information in irregular time series. Last but not least, we also investigate the scaling dynamics of TSFMs, which shows that model performance in general scales with model size.

Taken together, we demonstrate both the capabilities and the boundaries of TSFM zero-shot generalization on astrophysical time series with a comprehensive benchmark and analysis. Furthermore, we also provide additional meaningful insights beyond zero-shot in training better TSFMs on astrophysics data and irregular time series. By releasing all data, code, and model wrappers, `StarEmbed` serves to forge the bridge between astronomy and AI for time series fields and to provide the community with a benchmark for foundation model advances on challenging time-series data.

## Acknowledgments

We gratefully acknowledge the support of the NSF-Simons AI-Institute for the Sky (SkAI) via grants NSF AST-2421845 and Simons Foundation MPS-AI-00010513.

Dennis Wu is supported by the Cognitive Science Advanced Research Fellowship.

Han Liu is partially supported by NIH R01LM1372201, NSF AST-2421845, Simons Foundation MPS-AI-00010513, AbbVie, Dolby and Chan Zuckerberg Biohub Chicago Spoke Award.

Zwicky Transient Facility access for all authors was supported by Northwestern University and the Center for Interdisciplinary Exploration and Research in Astrophysics (CIERA). A.A.M. is supported by DoE award #DE-SC0025599. A.A.M. is also supported by Cottrell Scholar Award #CS-CSA-2025-059 from Research Corporation for Science Advancement. N.R. is supported by NSF award #2421845 and a Northwestern University Presidential Fellowship award.

This research has made use of NASA's Astrophysics Data System.

Based on observations obtained with the Samuel Oschin Telescope 48-inch and the 60-inch Telescope at the Palomar Observatory as part of the Zwicky Transient Facility project. ZTF is supported by the National Science Foundation under Grants No. AST-1440341, AST-2034437, and currently Award #2407588. ZTF receives additional funding from the ZTF partnership. Current members include Caltech, USA; Caltech/IPAC, USA; University of Maryland, USA; University of California, Berkeley, USA; University of Wisconsin at Milwaukee, USA; Cornell University, USA; Drexel University, USA; University of North Carolina at Chapel Hill, USA; Institute of Science and Technology, Austria; National Central University, Taiwan, and OKC, University of Stockholm, Sweden. Operations are conducted by Caltech's Optical Observatory (COO), Caltech/IPAC, and the University of Washington at Seattle, USA.

The CSS survey is funded by the National Aeronautics and Space Administration under Grant No. NNG05GF22G issued through the Science Mission Directorate Near-Earth Objects Observations Program. The CRTS survey is supported by the U.S. National Science Foundation under grants AST-0909182 and AST-1313422.

This research was supported in part through the computational resources and staff contributions provided for the Quest high performance computing facility at Northwestern University which is jointly supported by the Office of the Provost, the Office for Research, and Northwestern University Information Technology.

This research used both the DeltaAI advanced computing and data resource, which is supported by the National Science Foundation (award OAC 2320345) and the State of Illinois, and the Delta advanced computing and data resource which is supported by the National Science Foundation (award OAC 2005572) and the State of Illinois.. Delta and DeltaAI are joint efforts of the University of Illinois Urbana-Champaign and its National Center for Supercomputing Applications.

This work was completed in part at the NCSA Open Hackathon, part of the Open Hackathons program. The authors would like to acknowledge OpenACC-Standard.org for their support.

## Impact Statement

This paper introduces `StarEmbed`, a benchmark for evaluating time series foundation models on astronomical observations of variable stars. Our goal is to advance scientific understanding of how general time series foundation models perform on complex real-world time series data. While improved machine learning methods have broad societal implications, we do not identify any specific ethical concerns or negative impacts arising directly from this benchmarking study beyond those commonly associated with machine learning research.

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

# Appendix

## A. Additional Zero-shot and Fine-tuning Experiments

### A.1. Time Series Foundation Models for Classification

In this section, we extend our benchmark to include two TSFMs designed for classification.

**Mantis+ and MantisV2.** `Mantis+` and `MantisV2` (Feofanov et al., 2026) are two pre-trained TSFMs designed specifically for time series classification. Both share the same architecture: a 3-branch tokenizer followed by a 6-layer Transformer encoder. The three branches apply a 1D convolution to the raw signal, a convolution to the first-order temporal difference, and a per-patch mean/std encoding, respectively. They are pre-trained with a contrastive self-supervised objective on 2M synthetic time series. The two models differ only in their internal modules. `Mantis+` uses sinusoidal positional encoding, whereas `MantisV2` uses rotary positional encoding together with RMSNorm and SwiGLU activations. As a result, `MantisV2` has a more compact backbone (4.19M vs. 8.91M parameters). The contrastive objective on synthetic data is designed to produce embeddings that transfer well to unseen domains.

**Experiments Settings and Hyperparameters.** For zero-shot embedding, we follow the same per-band-then-concat pipeline used for the other TSFMs in our paper, with a context window of $192 = 32 \times 6$ (the largest multiple of 32 below 200, as required by `Mantis`'s tokenizer) and mean-pooled tokens from the third Transformer layer following from (Feofanov et al., 2026). For fine-tuning, we unfreeze the entire encoder and append the same 3-layer MLP head used in our zero-shot MLP benchmark. The $g$ and $r$ bands are fed as two input channels into the final Transformer layer. We use AdamW under a cosine learning-rate schedule, using the same class-weighted cross-entropy loss as the MLP benchmark. Hyperparameters (batch size, learning rate, dropout) are tuned with the same grid search protocol as the zero-shot MLP. We then run the best configuration with 10 seeds and report the mean and standard deviation in Table 10.

**Results.** Tables 8 and 9 contains the zero-shot clustering and classification results, respectively. Table 10 compares both the zero-shot and the fine-tuned classification results. In Table 9, both `Mantis+` and `MantisV2` perform on par with `Moirai` but score below `Chronos` and the hand-crafted feature baseline, which we attribute to their patch-based tokenization. For Table 10, we observe that fine-tuning improves both models substantially over their frozen-encoder baselines: with the same MLP head, full fine-tuning raises macro-F1 from $0.570$ to $0.638$ for `MantisV2` and from $0.600$ to $0.646$ for `Mantis+`. Even with this gain, fine-tuned Mantis remains below the hand-crafted feature baseline (macro-F1 $0.723$).

*Table 8.* Unsupervised clustering results for classification-specific TSFMs. Both Mantis models exhibit weaker clustering performance compared to forecasting-based TSFMs.

| Model | NMI (K-Means) | ARI (K-Means) | F1 (K-Means) | NMI (Ward) | ARI (Ward) | F1 (Ward) |
|---|---|---|---|---|---|---|
| `Astromer-1` | 0.0041(0.0001) | 0.0017(0.0011) | 0.1660(0.0014) | 0.0041 | 0.0001 | 0.1652 |
| `Astromer-2` | 0.0082(0.0010) | 0.0192(0.0078) | 0.1590(0.0042) | 0.0091 | 0.0310 | 0.1600 |
| `Moirai` | 0.1749(0.0017) | 0.0981(0.0028) | 0.2831(0.0034) | 0.1476 | 0.0828 | 0.2612 |
| `Moirai-2` | 0.2017(0.0019) | 0.0967(0.0060) | 0.3007(0.0095) | 0.1850 | 0.0603 | 0.2655 |
| `Chronos` | 0.2374(0.0082) | **0.1596(0.0029)** | 0.3110(0.0362) | 0.1890 | 0.1217 | **0.3671** |
| `Chronos-Bolt` | 0.2120(0.0033) | 0.1306(0.0125) | 0.3128(0.0027) | 0.2273 | **0.1553** | 0.3662 |
| `Time-MoE` | 0.2069(0.0108) | 0.0913(0.0090) | 0.3101(0.0227) | 0.1946 | 0.0944 | 0.3385 |
| Hand-crafted Features | **0.2700(0.0058)** | 0.1197(0.0092) | **0.3960(0.0271)** | **0.2508** | 0.1319 | 0.3323 |
| `MantisV2` | 0.0566(0.0006) | 0.0096(0.0032) | 0.1679(0.0064) | 0.0838 | 0.0346 | 0.2276 |
| `Mantis+` | 0.1012(0.0137) | 0.0619(0.0068) | 0.2217(0.0183) | 0.1172 | 0.0568 | 0.2117 |

### A.2. Fine-tuning Chronos

This subsection expands on the fine-tuning study in Section 5.6. We organize the experiments along two axes: (i) *fine-tuning objective* (forecasting vs. classification, Table 11), and (ii) *parameter-efficient/partial fine-tuning* (full fine-tuning vs. LoRA vs. LayerNorm-only, with either a linear or an MLP classification head, Tables 6 and 12).

#### A.2.1. CLASSIFICATION VS. FORECASTING FINE-TUNING

**Experiments Settings and Hyperparameters.** We compare two fine-tuning objectives for `Chronos-tiny`.

- **Chronos-FT_fcst.** We fine-tune on ZTF light curves with the Chronos forecasting objective (context=512, predic-

*Table 9.* Classification results including classification-specific TSFMs. We use frozen embeddings from layer 3 follow the paper setting. They show layer 3 yields the best feature-extraction performance. Other setting for embedding generation is the same as in our paper. `MantisV2` and `Mantis+` have comparable performance as `Moirai`. Unlike the point-wise tokenization of `Chronos`/`Time-MoE`, `Mantis` family uses patching like `Moirai`. We hypothesize this limit fine-grained temporal resolution on irregularly sampled light curves. See Figure 8 for more evidence. (Each metrics is averaged separately over 10 runs. Best results in **bold**, second-best underlined.)

| Classifier | Metric | Moirai | Moirai-2 | Chronos | Chronos-Bolt | Time-MoE | HC features | MantisV2 | Mantis+ |
|---|---|---|---|---|---|---|---|---|---|
| *k*-NN | Accuracy | 0.809 | 0.815 | 0.857 | 0.807 | 0.825 | **0.881** | 0.814 | 0.820 |
| | Precision | 0.662 | 0.641 | 0.799 | 0.647 | 0.724 | **0.818** | 0.657 | 0.667 |
| | Recall | 0.509 | 0.546 | 0.623 | 0.542 | 0.560 | **0.661** | 0.501 | 0.509 |
| | F1 | 0.554 | 0.561 | 0.672 | 0.570 | 0.595 | **0.712** | 0.550 | 0.557 |
| logistic | Accuracy | 0.705 | 0.709 | 0.750 | 0.709 | 0.740 | **0.838** | 0.743 | 0.739 |
| | Precision | 0.544 | 0.538 | 0.575 | 0.549 | 0.559 | **0.663** | 0.568 | 0.555 |
| | Recall | 0.680 | 0.678 | 0.730 | 0.676 | 0.694 | **0.854** | 0.711 | 0.737 |
| | F1 | 0.579 | 0.577 | 0.617 | 0.580 | 0.597 | **0.714** | 0.603 | 0.595 |
| RF | Accuracy | 0.823 (0.001) | 0.820 (0.001) | 0.862 (0.000) | 0.826 (0.001) | 0.839 (0.001) | **0.920 (0.001)** | 0.840 (0.001) | 0.845 (0.001) |
| | Precision | 0.716 (0.007) | 0.697 (0.010) | 0.750 (0.056) | 0.707 (0.002) | 0.693 (0.004) | **0.866 (0.003)** | 0.736 (0.003) | 0.731 (0.004) |
| | Recall | 0.514 (0.002) | 0.520 (0.004) | 0.597 (0.002) | 0.548 (0.001) | 0.561 (0.003) | **0.773 (0.004)** | 0.536 (0.002) | 0.533 (0.003) |
| | F1 | 0.557 (0.002) | 0.556 (0.003) | 0.638 (0.002) | 0.582 (0.001) | 0.598 (0.003) | **0.804 (0.003)** | 0.588 (0.002) | 0.585 (0.003) |
| MLP | Accuracy | 0.717 (0.031) | 0.735 (0.019) | 0.783 (0.022) | 0.721 (0.022) | 0.758 (0.025) | **0.833 (0.022)** | 0.686 (0.036) | 0.731 (0.034) |
| | Precision | 0.546 (0.022) | 0.560 (0.021) | 0.589 (0.025) | 0.553 (0.026) | 0.579 (0.024) | **0.672 (0.025)** | 0.551 (0.029) | 0.568 (0.019) |
| | Recall | 0.722 (0.006) | 0.717 (0.006) | 0.758 (0.006) | 0.696 (0.013) | 0.741 (0.010) | **0.851 (0.009)** | 0.714 (0.009) | 0.728 (0.010) |
| | F1 | 0.594 (0.019) | 0.603 (0.015) | 0.643 (0.023) | 0.589 (0.015) | 0.627 (0.019) | **0.723 (0.027)** | 0.570 (0.019) | 0.600 (0.021) |

*Table 10.* Comparison of zero-shot vs. fine-tuned classification for Mantis models. We append the same MLP head to the final layer of Mantis. The loss function is also the same weighted loss we use in MLP training. Hyperparameters were selected via the same grid search as the benchmark MLP. Best configuration: BS=256, LR=$10^{-5}$, dropout=0.1 for `MantisV2`; BS=256, LR=$10^{-5}$, dropout=0.0 for `Mantis+`. Full fine-tuning yields improvements, but still underperforms hand-crafted features. (Each metric is averaged separately over 10 runs. Best results in **bold**, second-best underlined.)

| | Zero-shot | | Fine-tuned | | |
|---|---|---|---|---|---|
| **Metric** | **MantisV2** | **Mantis+** | **MantisV2-FT** | **Mantis+-FT** | **HC features** |
| Accuracy | 0.686 (0.036) | 0.731 (0.034) | 0.773 (0.020) | 0.770 (0.018) | **0.833 (0.022)** |
| Precision | 0.551 (0.029) | 0.568 (0.019) | 0.605 (0.017) | 0.619 (0.016) | **0.672 (0.025)** |
| Recall | 0.714 (0.009) | 0.728 (0.010) | 0.780 (0.005) | 0.790 (0.014) | **0.851 (0.009)** |
| F1 | 0.570 (0.019) | 0.600 (0.021) | 0.638 (0.015) | 0.646 (0.018) | **0.723 (0.027)** |

tion=64), using the default Chronos fine-tuning configuration (LR=$10^{-3}$ with linear decay, and BS $= 64$). We finetune on $\sim$57k univariate time series from g+r bands) with early stopping setting patience 10 (eval every 200 steps).

- **Chronos-FT$_{cls}$.** We performs end-to-end classification fine-tuning. The $g$- and $r$-band light curves are fed through the shared encoder and the resulting 256-dim averaged embeddings are concatenated, then passed to the same MLP head and weighted cross-entropy loss as in our benchmark. Hyperparameters are selected via the same grid search as the benchmark MLP (BS $\in \{32, 64, 128, 256\}$, LR $\in \{10^{-3}, 10^{-4}, 10^{-5}\}$, dropout $\in \{0.0, 0.1\}$); best configuration: BS=32, LR=$10^{-4}$, dropout=0.1.

### A.2.2. FULL VS. PEFT VS. PARTIAL FINE-TUNING

**Experiments Settings and Hyperparameters.** Beside full fine-tuning, we further compare it with two other adaptation modes: LoRA ($r$=8, adapting the Q and V projections), and partial fine-tuning (only tuning the LayerNorms, following Chen et al. (2025)). Same as full fine-tuning, we test both forecasting and classification objectives.

- **Classification.** We pair each mode with either a linear head or an MLP head. Hyperparameters are selected via grid search over LR $\in \{10^{-3}, 10^{-4}, 10^{-5}\}$ and BS $\in \{32, 128, 256\}$, choosing the configuration with the highest macro-F1. Training uses early stopping with patience 5 on validation loss. All metrics are reported over 10 random seeds.

- **Forecasting.** HPO over LR $\in \{10^{-4}, 5\times10^{-4}, 10^{-3}\}$ and effective BS $\in \{32, 64, 128\}$ (gradient accumulation steps $= 2$), selecting the best checkpoint by forecasting validation loss (early stopping with patience 10). The downstream classifiers ($k$-NN with $k$=5 and logistic regression) are trained on the embeddings from each best checkpoint.

*Table 11.* Benchmark supervised classification including the two Chronos fine-tuning variants (`Chronos-FT_fcst` and `Chronos-FT_cls`). Best in **bold**, second-best underlined. `Chronos-FT_cls` improves 3 of 4 MLP metrics over zero-shot Chronos, while `Chronos-FT_fcst` does not improve downstream classification and is slightly worse than zero-shot Chronos. Since `Chronos-FT_cls` jointly trains a classification head with the encoder rather than producing reusable embeddings, it only applies to the MLP row. For RF and MLP, each metrics is averaged separately over 10 runs.

| Classifier | Metric | Moirai | Moirai-2 | Chronos-Bolt | Time-MoE | HC features | Chronos | Chronos-FT_fcst | Chronos-FT_cls |
|---|---|---|---|---|---|---|---|---|---|
| *k*-NN | Accuracy | 0.809 | 0.815 | 0.807 | 0.825 | **0.881** | 0.857 | 0.850 | — |
| | Precision | 0.662 | 0.641 | 0.647 | 0.724 | **0.818** | 0.799 | 0.712 | — |
| | Recall | 0.509 | 0.546 | 0.542 | 0.560 | **0.661** | 0.623 | 0.600 | — |
| | F1 | 0.554 | 0.561 | 0.570 | 0.595 | **0.712** | 0.672 | 0.630 | — |
| logistic | Accuracy | 0.705 | 0.709 | 0.709 | 0.740 | **0.838** | 0.750 | 0.738 | — |
| | Precision | 0.544 | 0.538 | 0.549 | 0.559 | **0.663** | 0.575 | 0.549 | — |
| | Recall | 0.680 | 0.678 | 0.676 | 0.694 | **0.854** | 0.730 | 0.713 | — |
| | F1 | 0.579 | 0.577 | 0.580 | 0.597 | **0.714** | 0.617 | 0.593 | — |
| RF | Accuracy | 0.823 (0.001) | 0.820 (0.001) | 0.826 (0.001) | 0.839 (0.001) | **0.920 (0.001)** | 0.862 (0.000) | 0.861 (0.001) | — |
| | Precision | 0.716 (0.007) | 0.697 (0.010) | 0.707 (0.002) | 0.693 (0.004) | **0.866 (0.003)** | 0.750 (0.056) | 0.733 (0.041) | — |
| | Recall | 0.514 (0.002) | 0.520 (0.004) | 0.548 (0.001) | 0.561 (0.003) | **0.773 (0.004)** | 0.597 (0.002) | 0.593 (0.003) | — |
| | F1 | 0.557 (0.002) | 0.556 (0.003) | 0.582 (0.001) | 0.598 (0.003) | **0.804 (0.003)** | 0.638 (0.002) | 0.628 (0.002) | — |
| MLP | Accuracy | 0.717 (0.031) | 0.735 (0.019) | 0.721 (0.022) | 0.758 (0.025) | **0.833 (0.022)** | 0.783 (0.022) | 0.746 (0.026) | 0.778 (0.029) |
| | Precision | 0.546 (0.022) | 0.560 (0.021) | 0.553 (0.026) | 0.579 (0.024) | **0.672 (0.025)** | 0.589 (0.025) | 0.586 (0.024) | 0.613 (0.036) |
| | Recall | 0.722 (0.006) | 0.717 (0.006) | 0.696 (0.013) | 0.741 (0.010) | **0.851 (0.009)** | 0.758 (0.006) | 0.739 (0.009) | 0.766 (0.005) |
| | F1 | 0.594 (0.019) | 0.603 (0.015) | 0.589 (0.015) | 0.627 (0.019) | **0.723 (0.027)** | 0.643 (0.023) | 0.627 (0.020) | 0.659 (0.027) |

*Table 12.* Classification fine-tuning results for Chronos-tiny on ZTF light curves with an **MLP head**. Same setup as Table 6. Extends the condensed version reported as Table 5 in the main text. Bold indicates the best in each column.

| Mode | Trainable Params | Macro F1 | Accuracy | Macro Prec | Macro Rec |
|---|---|---|---|---|---|
| Full FT | 9.6M (100%) | $0.659 \pm 0.026$ | **$0.778 \pm 0.028$** | **$0.613 \pm 0.034$** | $0.766 \pm 0.005$ |
| LoRA | 1.3M (13.2%) | **$0.661 \pm 0.015$** | $0.772 \pm 0.020$ | **$0.613 \pm 0.018$** | **$0.767 \pm 0.006$** |
| LayerNorm | 1.2M (12.4%) | $0.648 \pm 0.016$ | $0.774 \pm 0.023$ | $0.605 \pm 0.023$ | $0.766 \pm 0.006$ |

**Result.** See Tables 6, 12 and 13.

### A.3. Effect of Model Scale on Zero-Shot Performance

**Experiments Settings and Hyperparameters.** We evaluate the full range of publicly available model sizes for both Chronos (Tiny/Mini/Small/Base/Large, 8M–710M parameters) and Moirai (Small/Base/Large, 14M–311M parameters) to test whether larger models narrow the gap to hand-crafted features. For each model size we extract zero-shot embeddings using the same protocol as our main benchmark and evaluate two deterministic downstream classifiers, *k*-NN and logistic regression. Hand-crafted features serve as the reference baseline.

**Results.** See Table 14 and Figure 6.

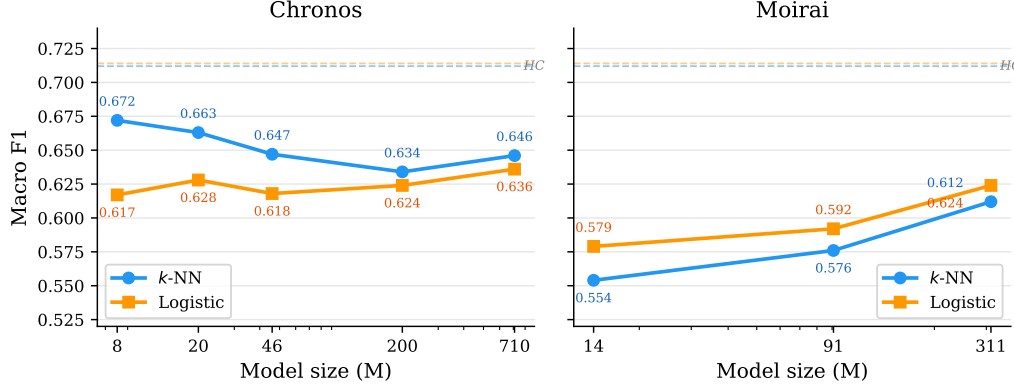

*Figure 6.* F1 score trend with increasing model size and hidden dimension.

*Table 13.* Forecasting PEFT/partial fine-tuning of Chronos-tiny on ZTF light curves, evaluated via downstream classification on the resulting embeddings. Bold indicates the best in each row. The result aligns with the full fine-tuning results in Table 11. The downstream classification on embeddings from the forecasting fine-tuned models still underperforms the zero-shot baseline. These confirm that the issue lies in the forecasting objective itself, which is unsuitable for a model that does not treat the irregular sampling of our time series properly (see Figure 4). Also see Appendix A.5 for possible improvements on this issue.

| Classifier | Metric | Zero-shot | Full FT | LoRA | LayerNorm |
|---|---|---|---|---|---|
| *k*-NN | Accuracy | 0.857 | 0.849 | **0.859** | 0.852 |
| | Precision | **0.799** | 0.735 | 0.783 | 0.762 |
| | Recall | 0.623 | 0.603 | **0.628** | 0.618 |
| | F1 | **0.672** | 0.632 | **0.672** | 0.654 |
| Logistic | Accuracy | **0.750** | 0.740 | **0.750** | 0.743 |
| | Precision | **0.575** | 0.564 | 0.569 | 0.568 |
| | Recall | **0.730** | 0.711 | 0.724 | 0.725 |
| | F1 | **0.617** | 0.601 | 0.610 | 0.608 |

*Table 14.* Effect of model scale on downstream classification. Moirai benefits from scaling on both classifiers, and Chronos improves on logistic regression with scale. The gap to hand-crafted features narrows with increasing model size, though a meaningful gap remains even at the largest available scales.

| Classifier | Metric | Chronos | | | | | Moirai | | | HC features |
|---|---|---|---|---|---|---|---|---|---|---|
| | | Tiny | Mini | Small | Base | Large | Small | Base | Large | — |
| | # Params | 8M | 20M | 46M | 200M | 710M | 14M | 91M | 311M | — |
| | Emb. dim | 512 | 768 | 1024 | 1536 | 2048 | 768 | 1536 | 2048 | 138 |
| *k*-NN | Accuracy | 0.857 | 0.859 | 0.848 | 0.849 | 0.850 | 0.809 | 0.824 | 0.849 | 0.881 |
| | Precision | 0.799 | 0.769 | 0.775 | 0.751 | 0.785 | 0.662 | 0.709 | 0.722 | 0.818 |
| | Recall | 0.623 | 0.617 | 0.597 | 0.597 | 0.612 | 0.509 | 0.546 | 0.590 | 0.661 |
| | F1 | 0.672 | 0.663 | 0.647 | 0.634 | 0.646 | 0.554 | 0.576 | 0.612 | 0.712 |
| Logistic | Accuracy | 0.750 | 0.766 | 0.764 | 0.783 | 0.794 | 0.705 | 0.751 | 0.780 | 0.838 |
| | Precision | 0.575 | 0.587 | 0.580 | 0.588 | 0.602 | 0.544 | 0.558 | 0.598 | 0.663 |
| | Recall | 0.730 | 0.731 | 0.706 | 0.695 | 0.698 | 0.680 | 0.658 | 0.673 | 0.854 |
| | F1 | 0.617 | 0.628 | 0.618 | 0.624 | 0.636 | 0.579 | 0.592 | 0.624 | 0.714 |

## A.4. Zero-Shot OOD Detection with Local Outlier Factor

**Experiments Settings and Hyperparameters.** To test the robustness of our OOD results, we introduce additional evaluations in this section. To complement the multi-class isolation forest method described in Section 5.3, we also test our embedding using Local Outlier Factor (LOF). We choose LOF because it imposes a different inductive bias than the isolation forest (IF) method. While IF is tree/partition-based, LOF is density-based and tests a different aspect of the embedding geometry. Our results in Table 15 confirm that, while the absolute purity values differ across IF and LOF, the relative trend is consistent across both OOD tests with the Chronos models performing the best on the OOD tests.

## A.5. TSFM Pre-training: Modified RoPE for Irregular Time Series

This subsection provides the full experimental setup and additional qualitative results supporting the modified-RoPE study reported in the main text (Table 7).

**Experiments Settings and Hyperparameters.** **Model.** We use `Moirai` as the base TSFM because it natively supports RoPE and input covariates. *Variants.* We evaluate three architectural variants of the positional treatment: (1) *RoPE + True timestamp* — replacing sequential position indices with actual observation timestamps; (2) $\Delta t$ *as covariate* — appending inter-observation time gaps as an additional input variate; (3) *Vanilla* — neither modification. **Data.** Each sample follows $x(t) = A\sin(2\pi ft + \varphi) + \varepsilon$ with $f \sim \mathcal{U}(0.5, 3.0)$, $A \sim \mathcal{U}(0.5, 2.0)$, $\varphi \sim \mathcal{U}(0, 2\pi)$, $\varepsilon \sim \mathcal{N}(0, 0.05)$. Each time series contains 160 observations spanning $t \in [0, 10]$. The context window is $t < 7$ (~112 points), the prediction window $t \geq 7$ (~48 points), and the train/val/test split is $1,000/200/200$. **Training.** LR $= 5 \times 10^{-4}$ with cosine decay, batch size 128, early stopping with patience 20. **Metrics.** Pearson correlation for shape recovery, plus $R^2$, MSE, and MAE for point-wise

*Table 15.* Purity of top N-percentile of sources selected by out-of-distribution source detection using Local Outlier Factor (LOF). We choose LOF because it has a different inductive bias from isolation forest. The best results are highlighted in **bold**, and the second-best are underlined. We find that the chronos-bolt model continues to perform the best with chronos being a close second. With the LOF method, we also report much stronger performance for the best TSFM models when compared to the hand-crafted features, further strengthening the case to use TSFMs for OOD detection.

|  | **Local Outlier Factor** | | |
| --- | --- | --- | --- |
|  | **Top 1%** | **Top 5%** | **Top 10%** |
| Astromer-1 | 0.108 | 0.167 | 0.203 |
| Astromer-2 | 0.236 | 0.220 | 0.187 |
| Moirai | 0.281 | 0.232 | 0.184 |
| Moirai-2 | 0.079 | 0.122 | 0.126 |
| Chronos | **0.989** | **0.998** | 0.992 |
| Chronos-Bolt | **0.989** | 0.996 | **0.997** |
| Time-MoE | 0.157 | 0.198 | 0.179 |
| Random | 0.116 | 0.116 | 0.116 |
| HC features | 0.236 | 0.229 | 0.210 |

prediction accuracy.

**Result.** See Table 16 and prediction examples in Figure 7.

*Table 16.* Forecasting irregular sinusoidal time series with three architectural variants of MOIRAI's positional treatment. RoPE + True timestamp outperforms $\Delta t$ as covariate and Vanilla, indicating that preserving the shape of the irregular time series through positional encoding is the key. Extends the condensed comparison in Table 7; see prediction examples in Figure 7.

| Variant | $R^2 \uparrow$ | Pearson $\uparrow$ | MSE $\downarrow$ | MAE $\downarrow$ |
| --- | --- | --- | --- | --- |
| RoPE + True timestamp | $\mathbf{0.188 \pm 0.102}$ | $\mathbf{0.351 \pm 0.114}$ | $\mathbf{0.041 \pm 0.006}$ | $\mathbf{0.159 \pm 0.017}$ |
| $\Delta t$ as covariate | $0.044 \pm 0.035$ | $0.277 \pm 0.058$ | $0.049 \pm 0.002$ | $0.177 \pm 0.006$ |
| Vanilla | $0.016 \pm 0.039$ | $0.202 \pm 0.098$ | $0.050 \pm 0.002$ | $0.183 \pm 0.007$ |

Ranked by rope_only R² (rank 10,20,30,40,50,60,70,80): rope_only vs dt_only vs vanilla

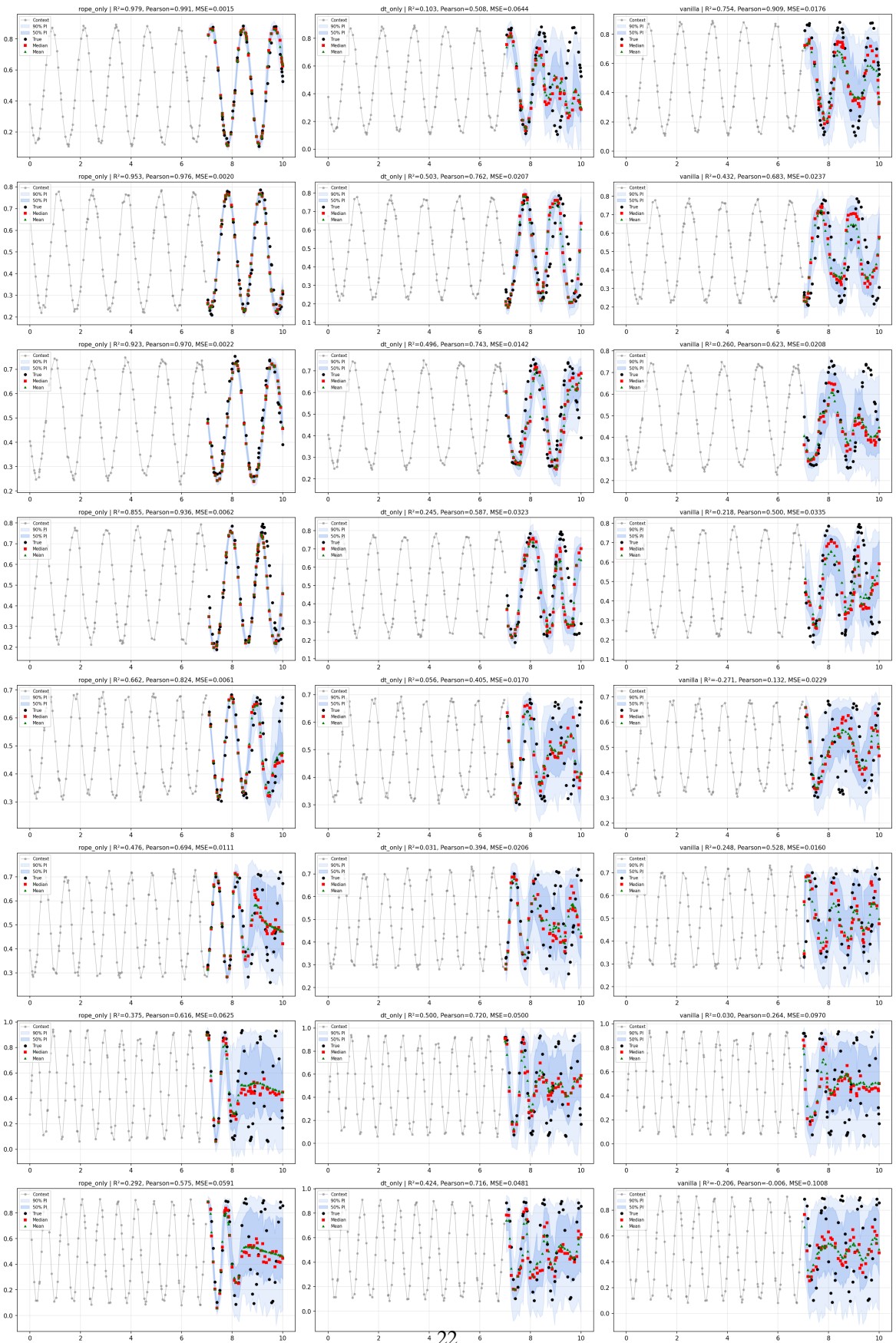

*Figure 7.* Predictions sorted by RoPE-only $R^2$ (ranks 10–80 out of 200), spanning from successful to failed predictions.

# B. Extended Analysis of Results and Experiment Details

## B.1. Attention Analysis

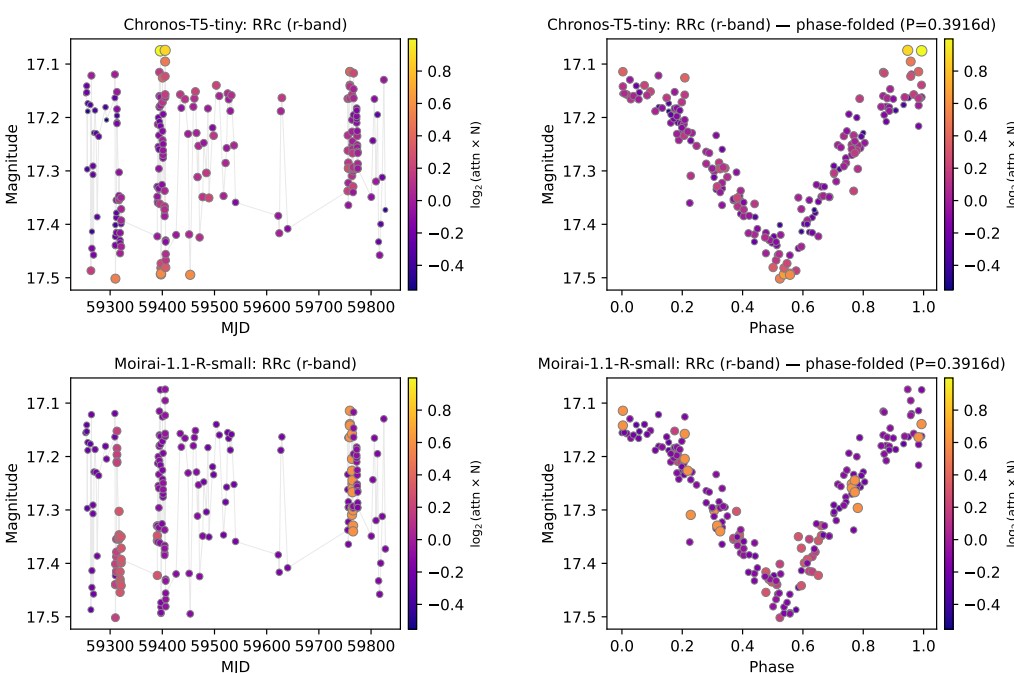

*Figure 8.* Chronos concentrates attention on brightness extrema in the phase-folded light curve, despite receiving only raw, irregularly sampled magnitudes without period information (we train a Chronos-embedding linear regressor and get $R^2 < 0.13$ for the period feature). **This indicates Chronos succeeds in attending to morphologically distinctive segments in the raw light curve (extreme values, rapid brightness changes) that coincide with physically meaningful phases.** Moirai, in contrast, distributes attention nearly uniformly or assigns high scores to non-distinctive segments. This stems from its patching mechanism: grouping consecutive observations into patches (16 time steps) conflates observations separated by irregular time gaps, destroying local morphological structure. See Figure 9 for other six classes.

**Attention Rollout Setup.** We visualize attention patterns of `Chronos-tiny` and `Moirai-small` via attention rollout (Abnar & Zuidema, 2020). At the $\ell$-th layer, the attention matrix $A_\ell$ is averaged across heads and blended with the identity matrix ($0.5 \cdot A_\ell + 0.5 \cdot I$) to account for residual connections. The blended matrices are then multiplied across all layers to yield one importance score $attn$ per observation. We report $\log_2(N \cdot attn)$, where zero indicates averaged attention (every observation gets equal attention), positive values indicate above-average attention, and negative values indicate below-average attention.

## B.2. TSFMs Embedding Visualization (UMAP)

We include the UMAP visualizations for the embeddings from each embedding model to provide more intuition regarding the embedding space. As shown by Figure 10, all time-series pre-trained models' embeddings show clear distinction and distribution of different clusters corresponding to different ground-truth classes. In comparison, as a baseline, the random embeddings show no clear clusters at all. `Astromer-1`'s embeddings and hand-crafted features do not show clear clusters either. `Astromer-2`'s embeddings show clearer cluster distribution but for some classes, the clusters are not distinctive from others either. These UMAP visualizations further demonstrate the promising potential of using time-series pre-trained models as light-curve embedding models.

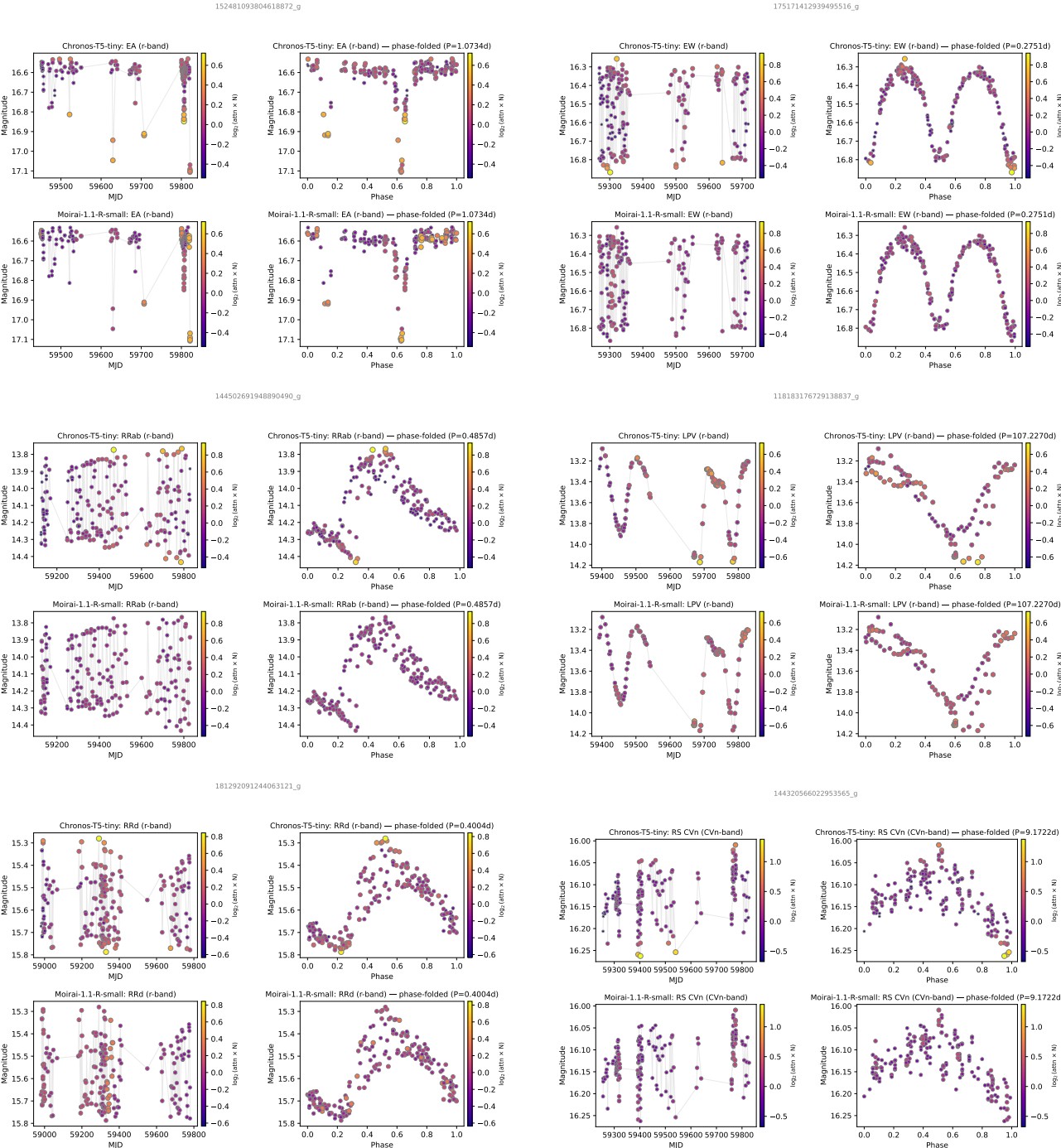

*Figure 9.* Attention rollout visualizations for the other six classes (EA, EW, RRab, LPV, RRd, RS CVn). In each subplot, the upper row is `Chronos`, and the lower row is `Moirai`.

### B.3. `Astromer` Embedding Analysis

We provide detailed analysis and additional experiments on the issue of the poor performance of `Astromer-1`. First, `Astromer-1` needs further fine-tuning on the downstream-task dataset to achieve good performance on the variable-star classification task, according to Donoso-Oliva et al. (2023). This is evidenced in their Fig. 11(a), which shows a clear increase of F1 score from ~0.25 to ~0.6 when fine-tuned on 20 to 500 variable stars per class of the MACHO dataset. A similar trend is observed for other datasets, including OGLE-III and ATLAS. `Astromer-2` (Donoso-Oliva et al., 2026) also shows improvement with fine-tuning, though its performance starts higher (around 0.65) even with just 20 samples per

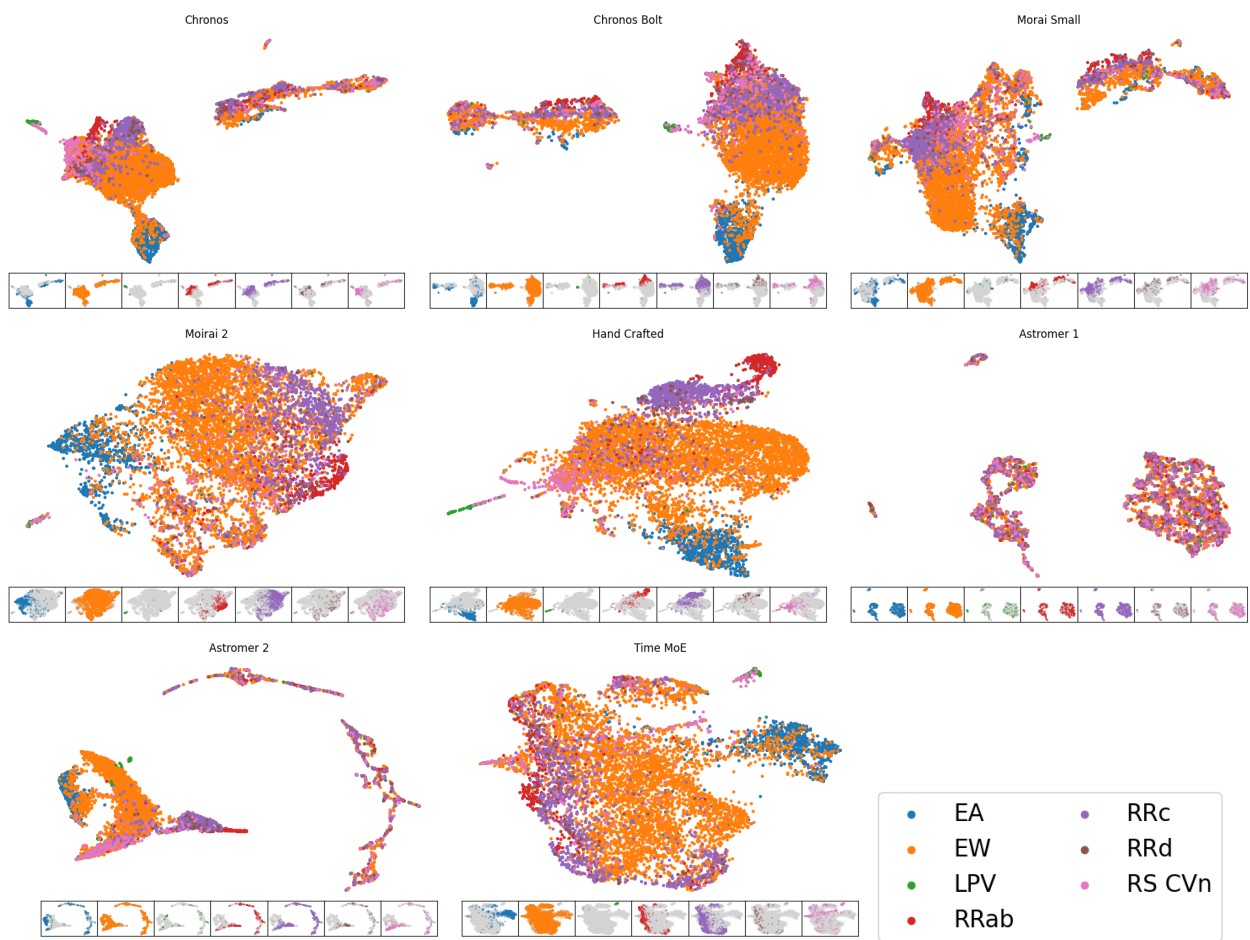

*Figure 10.* UMAP projections for each embedding model included in our analysis using the test set. Inset plots at the bottom of each figure show clustering of different classes.

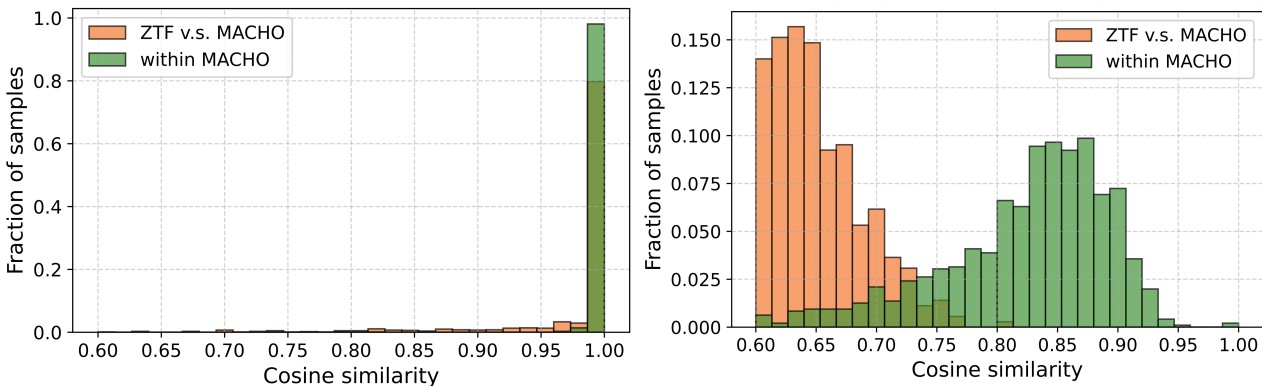

*Figure 11.* **Comparison of embedding alignment for `Astromer-1` (left) and `Chronos` (right).** The plots show the distribution of cosine similarities between light-curve embeddings, both within the MACHO survey (green) and across surveys (ZTF vs. MACHO, orange). `Astromer-1` exhibits embedding collapse, with cosine similarities approaching 1.0 no matter within- or cross-survey. This indicates that the model encodes little discriminative structure. In contrast, `Chronos` produces more meaningful embeddings: the wider distribution of cosine similarities preserves structural information, and the clear separation between within-survey and cross-survey pairs demonstrates its ability to capture the domain shift between datasets.

class. Please refer to Figures 11 and 12 of Donoso-Oliva et al. (2026) for details. These results indicate that `Astromer`'s low zero-shot performance in our benchmark is expected, as we intentionally evaluate the pre-trained checkpoints without any task-specific tuning.

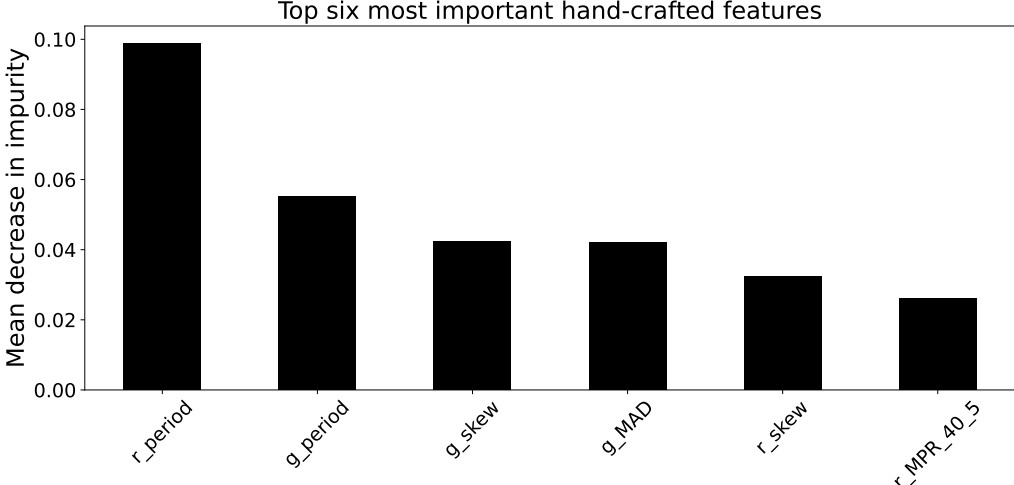

*Figure 12.* Top six most important hand-crafted features for random forest classification. The period of variability computed for either passband rank the highest. Other important metrics also include the skewness of the distribution of magnitudes, the median absolute deviation, and the magnitude percentage ratio for the 40th and 5th percentiles.

Second, we provide empirical analysis to show that `Astromer-1` embeddings collapse into similar directions. Specifically, we randomly sample 1000 pairs of ZTF–MACHO data in the r-band, and compute the cosine similarity of embeddings from two models, `Astromer-1` and `Chronos`. For comparison, we do the same for within-MACHO pairs. The result is in Figure 11. For `Astromer-1`, it is clear that its embeddings collapse to one direction. Even the 10th percentile of cosine similarity of embeddings within MACHO is 0.995, meaning every star is almost parallel to every other. Even under the survey-data shift, the 10th percentile is still 0.948. From the above we conclude that the embeddings of frozen `Astromer-1` encode very little discriminative structure. This explains why a downstream classifier has a hard time distinguishing different classes (low F1 score). For `Chronos` (right figure), the cosine similarity within MACHO (green) shows a wide range of angles, indicating the embeddings preserve class information. Furthermore, the embedding has a clear domain shift: the cosine similarity of ZTF–MACHO pairs (orange) is lower than that of within-MACHO pairs. Unlike frozen `Astromer-1`, `Chronos` does not collapse everything into a single direction. It still has room to spread out unseen patterns instead of forcing them into the old manifold.

### B.4. Importance Analysis on Hand-crafted Features

In this appendix we investigate the feature importance of the individual hand-crafted features to investigate the source of their excellent performance. We perform this analysis on the random forest trained for supervised classification with the hand-crafted features because it offers built-in functionality for quantitatively measuring the importance of each feature. The feature importance is computed using the mean decrease in impurity. In this framework, each time a feature is used to split a node, the associated reduction in the Gini impurity metric is recorded and weighted by the number of samples reaching that node. These weighted impurity decreases are then summed over all nodes and all trees in the ensemble, and finally normalized to yield a global, unit-scaled measure of how strongly each feature contributes to improving class separation within the forest.

Figure 12 shows the results of our feature importance analysis. We find that the period inferred from the $g$ and $r$ bands of the time series are the most important features driving the performance of the hand-crafted features. This aligns with expectations as numerous other studies performing variable star classification with tree-based classifiers come to a similar conclusion (Richards et al., 2011; Dubath et al., 2011; Kim & Bailer-Jones, 2016). We also find that the skewness, median absolute deviation, and the magnitude percentage ratio also rank very highly. These features encode information about the spread of the magnitude measurements, indirectly representing the morphology of the periodicity of the star. All features are described in Appendix C.

*Table 17.* Best hyper-parameters for each model with the MLP classifier

| Method | Hyperparameters | Training epochs |
|---|---|---|
| `Astromer-1` | `batch_size=32, learning_rate=0.0001, dropout=0.0` | 17 |
| `Astromer-2` | `batch_size=32, learning_rate=0.0001, dropout=0.0` | 23 |
| `Moirai` | `batch_size=64, learning_rate=0.001, dropout=0.0` | 11 |
| `Moirai-2` | `batch_size=32, learning_rate=0.0001, dropout=0.1` | 5 |
| `Chronos` | `batch_size=32, learning_rate=0.0001, dropout=0.0` | 26 |
| `Chronos-Bolt` | `batch_size=128, learning_rate=0.0001, dropout=0.1` | 17 |
| `Time-MoE` | `batch_size=256, learning_rate=0.0001, dropout=0.1` | 14 |
| Random Embeddings | `batch_size=128, learning_rate=0.0001, dropout=0.0` | 5 |
| Hand-crafted features | `batch_size=32, learning_rate=0.0001, dropout=0.1` | 30 |

*Table 18.* Best hyperparameters for each model with the random forest classifier

| Method | Hyper-parameters | Training time (s) |
|---|---|---|
| `Astromer-1` | `max_depth=10, min_samples_split=2, n_estimators=200` | 84 |
| `Astromer-2` | `max_depth=30, min_samples_split=10, n_estimators=500` | 456 |
| `Moirai` | `max_depth=None, min_samples_split=2, n_estimators=500` | 738 |
| `Moirai-2` | `max_depth=None, min_samples_split=2, n_estimators=100` | 138 |
| `Chronos` | `max_depth=None, min_samples_split=5, n_estimators=100` | 126 |
| `Chronos-Bolt` | `max_depth=30, min_samples_split=2, n_estimators=500` | 450 |
| `Time-MoE` | `max_depth=30, min_samples_split=5, n_estimators=500` | 252 |
| Random Embeddings | `max_depth=None, min_samples_split=2, n_estimators=100` | 198 |
| Hand-crafted features | `max_depth=None, min_samples_split=10, n_estimators=100` | 36.4 |

### B.5. Hyperparameters for Classification

We report the hyperparameter tuning process and summary for all classifiers in this section. To fairly compare the different embeddings, we conduct a hyperparameter search on each model when training the downstream MLP and random forest classifier. We use MLP with three hidden layers of sizes 1024, 512 and 256 and an output layer for class predictions. we search over batch size $B \in \{128, 256, 512, 1024\}$, learning rate $lr \in \{0.01, 0.001, 0.0001\}$ and dropout rate $\in \{0.0, 0.1\}$. Every hyperparameter triple runs once on NVIDIA H100 4 GPUs. The training process is at most 50 epochs, and stops early if the validation loss fails to improve for 3 epochs. In practice this training takes less than 30 epochs for all models before the early stopping is triggered. For random forest, we search over maximum depth of the tree $\in \{$None$, 10, 20, 30\}$, the minimum number of samples to split an internal node $\in \{2, 5, 10\}$, and number of estimator $\in \{100, 200, 500\}$. We summarize the hyperparameters of the MLP and random forest classifiers in Table 17 and 18. For linear classifier, since the current training set is relatively small, we use the `LogisticRegression` (L-BFGS) from Scikit-learn library (Buitinck et al., 2013), with `max_iter = 5000`, `class_weight = "balanced"`, and all other with default settings. L-BFGS is a deterministic full-batch method that converges to the global minimizer without learning rate tuning. For $k$-NN, we use `KNeighborsClassifier` from Scikit-learn with default settings. Since $k$-NN and the default solver of logistic regression are both deterministic, we only report the 1 run result.

### B.6. Computational Analysis

Here we compare the computational cost of the models we evaluate. To compare the computational cost for *inference*, we calculate the FLOPs of the forward pass of each model. The results are in Table 19. Model FLOPs are referenced from Table 2 in the main text. Table 20 lists the formulas used to compute FLOPs for each component, and Table 21 summarizes architectures, parameter counts, embedding dimensions and total FLOPs side-by-side.

*Table 19.* FLOPs per forward pass (batch size = 1), with $L$=200 tokens and patch size $P$=1 applied uniformly to all models. Each multiply–add is counted as 2 FLOPs. For encoder–decoder models (Chronos-tiny, Chronos-Bolt-tiny), attention linear and quadratic columns aggregate self-attention in both encoder and decoder plus cross-attention in the decoder; the FFN column aggregates both stacks. For Time-MoE-50M, only *active* FLOPs are reported (3 of 9 experts activated per layer); the total dense-equivalent would be $\approx 61$ GFLOPs. Quadratic attention FLOPs scale as $O(L^2 d)$; FFN FLOPs scale as $O(L \cdot d \cdot d_{\text{ff}})$. ASTROMER-1/2 already operated at $L$=200, $P$=1 and are unchanged. Under these conditions, Chronos-Bolt-tiny and Chronos-tiny share the same transformer backbone ($d$=256, $d_{\text{ff}}$=1024, 4 + 4 layers) and thus yield identical FLOPs; the sole original difference—input patching—is eliminated when $P$=1. Totals reflect main transformer operations only (attention + FFN); auxiliary components such as positional encodings and normalisation layers contribute negligibly.

| Model | Attention (MFLOPs) | | FFN (MFLOPs) | Total (GFLOPs) |
|---|---|---|---|---|
| | Linear | Quadratic | | |
| ASTROMER-1 | 209.7 | 41.0 | 52.4 | **0.30** |
| ASTROMER-2 | 629.1 | 123.0 | 157.3 | **0.91** |
| Chronos-tiny[†] | 1,258.2 | 246.0 | 1,677.8 | **3.18** |
| Chronos-Bolt-tiny[†] | 1,258.2 | 246.0 | 1,677.8 | **3.18** |
| Moirai-2.0-small | 1,415.7 | 184.3 | 1,887.4 | **3.49** |
| Moirai-small | 1,415.7 | 184.3 | 2,831.2 | **4.43** |
| Time-MoE-50M[‡] | 2,831.5 | 368.6 | 16,986.9 | **20.19** |

[†] Identical FLOPs: both models share the same backbone at $P$=1.

[‡] Active FLOPs only (3 of 9 experts per MoE layer).

*Table 20.* FLOPs formulas used for each component (1 multiply-add = 2 FLOPs). $N$ = number of layers, $L$ = sequence length, $d$ = model dimension, $d_{\text{ff}}$ = FFN width, $E_a$ = active experts per token.

| Component | FLOPs formula |
|---|---|
| Attn projections (Q, K, V, O) | $N \times 4 \times 2 \times L \times d^2$ |
| Attn scores + weighted sum | $N \times 2 \times 2 \times L^2 \times d$ |
| FFN (two linear layers) | $N \times 2 \times 2 \times L \times d \times d_{\text{ff}}$ |
| Cross-attn linear | $N \times (2L_{\text{dec}} + 2L_{\text{enc}}) \times 2 \times d^2$ |
| Cross-attn scores + weighted sum | $N \times 2 \times 2 \times L_{\text{dec}} \times L_{\text{enc}} \times d$ |
| MoE FFN (active) | $E_a \times N \times 2 \times 2 \times L \times d \times d_{\text{ff}}$ |

*Table 21.* Model architectures, scales, parameter counts, embedding dimensions and forward-pass FLOPs. FLOPs are reported in MFLOPs except Total FLOPs, which is in GFLOPs. FLOPs numbers are from Table 19.

| Model | Architecture family | Exact scale used | Parameter count | Emb. dim | Attn. Linear | Attn. Quad. | FFN | Total FLOPs |
|---|---|---|---|---|---|---|---|---|
| Astromer-1 | astronomy-specific transformer embedding model | Astromer-1 | 0.66M | 256 | 209.7 | 41.0 | 52.4 | 0.30 |
| Astromer-2 | astronomy-specific transformer embedding model | Astromer-2 | 5.4M | 256 | 629.1 | 123.0 | 157.3 | 0.91 |
| Moirai | masked encoder-based universal forecasting transformer | Moirai-small | 14M | 384 | 1415.7 | 184.3 | 2831.2 | 4.43 |
| Moirai-2 | decoder-only universal forecasting transformer | Moirai-2.0-small | 11.4M | 384 | 1415.7 | 184.3 | 1887.4 | 3.49 |
| Chronos | T5-style quantized time-series language model | Chronos-tiny | 8M | 256 | 1258.2 | 246.0 | 1677.8 | 3.18 |
| Chronos-Bolt | patch-based direct multi-step T5 encoder-decoder | Chronos-Bolt-tiny | 9M | 256 | 1258.2 | 246.0 | 1677.8 | 3.18 |
| Time-MoE | decoder-only mixture-of-experts TSFM | Time-MoE-50M | 50M activated / 113M total | 384 | 2831.5 | 368.6 | 16986.9 | 20.19 |
| Hand-crafted features | domain-engineered feature baseline | — | — | 138 features | — | — | — | — |

## C. Full List of Hand-crafted Features

We select hand-crafted features from the libraries of established software packages: FATS (Nun et al., 2015) and light_curve (Malanchev et al., 2021). Each feature described here is computed for each passband individually and the embeddings are formed by concatenating the feature lists of the $g$ and $r$ embeddings. Tables 23 and 22 show the full list of features and descriptions from FATS and light_curve, respectively.

*Table 22.* light_curve features and plain-language descriptions

| Feature | Intuitive one-sentence description |
| --- | --- |
| Amplitude | Half the peak-to-peak range—how far the light curve swings between brightest and faintest points. |
| AndersonDarlingNormal | Scores how strongly the magnitude distribution departs from an ideal bell curve. |
| BeyondNStd | Proportion of data points that sit more than $N$ standard deviations away from the mean, flagging outliers. |
| Cusum | Total vertical span of the running cumulative sum, revealing slow drifts or trends. |
| Eta | Von Neumann ratio: compares successive-point differences to overall scatter to catch rapid variability. |
| EtaE | Eta re-weighted by time gaps so uneven sampling doesn't skew the variability estimate. |
| InterPercentileRange($p$) | Distance between the $p$ and $(1 - p)$ quantiles—a robust width such as the IQR (when $p$=0.25). |
| Kurtosis | Indicates whether the distribution is more peaked or heavy-tailed than a normal curve. |
| LinearFit | Slope, error, and fit quality for a straight line that accounts for measurement uncertainties. |
| LinearTrend | Slope and error of a simple least-squares line that ignores the error bars. |
| MagnitudePercentageRatio | Ratio of inner to outer percentile widths, contrasting core spread with overall spread. |
| MaximumSlope | Steepest single-step change in magnitude per unit time between consecutive points. |
| Mean | Ordinary average magnitude. |
| Median | Mid-point magnitude that splits the data into equal halves. |
| MedianAbsoluteDeviation | Typical absolute distance from the median—a robust scatter measure. |
| MedianBufferRangePercentage | Fraction of points that fall inside a narrow buffer zone around the median. |
| OtsuSplit | Statistics describing the two groups produced by Otsu's automatic thresholding of magnitudes. |
| PercentAmplitude | Largest absolute deviation of any point from the median magnitude. |
| ReducedChi2 | Reduced $\chi^2$ showing how well the data match their (weighted) mean given the quoted errors. |
| Skew | Tells whether the distribution leans toward brighter or fainter extremes (positive or negative tail). |
| StandardDeviation | Classical root-mean-square scatter of the magnitudes. |
| StetsonK | Error-weighted "peakedness" measure that is robust to outliers in light-curve shape. |
| WeightedMean | Average magnitude that gives greater weight to points with smaller measurement errors. |

See here for a detailed description: https://github.com/light-curve/light-curve-python

## D. Astrophysical Description of Classes

Our dataset contains seven total classes of periodic variable stars: EW, EA, RRab, RRc, RRd, RS CVn, and LPV. Here, we provide a high-level astrophysical description of each of these classes, including each class' observational characteristics and

*Table 23.* FATS features and plain-language descriptions.

| Feature | Intuitive one-sentence description |
|---|---|
| PeriodLS | Best-fit period of the light curve using the Lomb–Scargle method. |
| Period_fit | The false alarm probability of the largest Lomb–Scargle periodogram value. |
| Psi_CS | The range of a cumulative sum metric computed on the phase-folded light curve. |
| Psi_eta | The variability index $\eta^e$ computed on the phase-folded light curve. |
| Autocor_length | The cross-correlation of the light curve with itself. |
| PairSlopeTrend | The fraction of increasing first differences subtracted from the fraction of decreasing first differences, computed on the 30 most recent magnitude measurements. |
| Freq{N}_harmonics_amplitude_{M} | Amplitude of the $M$th harmonic of the $N$th dominant frequency. |
| Freq{N}_harmonics_rel_phase_{M} | Relative phase of the $M$th harmonic of the $N$th dominant frequency. |
| CAR_sigma | Short-term variability amplitude in a continuous auto-regressive (CAR) model. |
| CAR_tau | Characteristic timescale of correlations in the CAR model. |
| CAR_mean | Long-term mean magnitude level in the CAR model. |

See here for a detailed description: http://isadoranun.github.io/tsfeat/FeaturesDocumentation.html

utility. In some cases, multiple classes are closely related so we describe them together. We also include descriptions of the classes which, due to their rarity, are considered out-of-distribution in this work: $\beta$-Lyrae, Blazhko, Anomalous Cepheids, Cepheid-II, HADS, LADS, ELL, Hump, PCEB, and EAup.

### D.1. Eclipsing Binaries (EW, EA)

Eclipsing binary stars are pairs of stars orbiting each other and aligned with the observer in such a way that either star periodically blocks the light from the other. When neither star is eclipsed, the system is at maximum brightness, but when one star is eclipsed by the other, the total flux received from the system is suppressed, giving the binary star system periodic light curve behavior. This type of variability is described as extrinsic because it is not due to astrophysical properties of the stars themselves. Sub-categorization of eclipsing binaries is based on the configuration of the stars in the pair.

EW-type eclipsing binaries (also called W Ursae Majoris-type after the original EW system) are contact binaries. In this case, the two stars, typically dwarf stars, share a common outer envelope which entirely encapsulates them. This common envelope allows for the exchange of mass and energy between the pair, equilibrating their temperature. Their light curves exhibit constant and smooth variations where the dips from either star being eclipsed are of similar or identical depth. EW-type variables typically have short periods ($0.2 \lesssim P$ [d] $\lesssim 0.5$), with a notably unsolved period cut-off at $\sim 0.2$ days (Rucinski, 2007; Drake et al., 2014). These systems are astrophysically valuable, in part, because they are expected to emit gravitational waves due to their tight orbits, and they also have the possibility of merging and triggering transient events (Tylenda et al., 2011).

EA-type eclipsing binaries (also called Algol-type binaries after the original EA system) are detached binaries. In this case, the two stars are not in contact and thus can have different temperatures and more varied orbits, manifesting as different light curve properties. The ellipticity of the orbit and the brightnesses of the stars affect the spacing and depths of the brightness dips. Multiple additional factors can affect their light curves, e.g., the presence of an accretion disk. EA systems tend to have longer periods than EW systems due to their wider orbits: $0.3 \lesssim P$ [d] $\lesssim 100$. EA systems are key for studying binary stellar evolution and populations, especially the exchange of mass between stars in a binary.

### D.2. Active Binaries (RS CVn)

RS Canum Venaticorum (RS CVn) stars are also stars in binary systems and are characterized by one of the stars exhibiting large magnetic spots on its surface. This manifests as observable variability as the RS CVn stars also have rapid rotational velocities. The resulting light curve affect also depends on the difference between the star's rotational period and the systems orbital period, and most RS CVn systems are tidally locked, meaning the two periods are closely matched. Some RS CVn are also eclipsing binaries, so their light curves can also show variability due to eclipses. Their periods tend to be

$3 \lesssim P$ [d] $\lesssim 14$. RS CVn systems serve as extreme testbeds for studying stellar magnetic phenomena and evolution.

### D.3. RR Lyrae (RRab, RRc, RRd)

RR Lyrae stars are low-mass stars exhibiting pulsations, cyclically expanding and contracting radially due to internal changes in opacity. Because RR Lyrae occur with only a small range of intrinsic brightnesses, their distances can be easily measured from their observed brightness. Among other utilities, this allows RR Lyrae to be used for measuring distances within the Milky Way and to nearby Galaxies. Their periods are also related to their chemical composition, so they can provide crucial information about Galactic structure and formation.

RR Lyrae can occur in different pulsation modes which define the various RR Lyrae subclasses. RRab stars pulsate in the fundamental mode; RRc in the first-overtone; and RRd are double-mode pulsators. RRab stars have light curves with a rapid brightening episode followed by a gradual fading, producing a sawtooth-like pattern. RRab typically have periods $0.4 \lesssim P$ [d] $\lesssim 1.0$. RRc stars exhibit sinusoid-like variability with typical periods of $0.2 \lesssim P$ [d] $\lesssim 0.5$. They also tend to exhibit a constant, slow drift in their periods. RRd stars pulsate in both the fundamental mode and the first-overtone and thus show a combination of two periodic signals in their light curves, typically with the first-overtone dominating.

### D.4. Long Period Variables (LPVs)

LPVs are giant stars exhibiting pulsations with periods of $3 \lesssim P$ [d] $\lesssim 1000$. They include multiple subtypes each with different period-luminosity relations but we, as in (Drake et al., 2014), consider these a single class. They pulsate with a similar mechanism to the RR Lyrae but have dramatically larger radii. Their outer layers are not very tightly bound to the rest of the star, which can lead to the star expunging mass and polluting the surrounding environment with gas. Thus, studying LPVs provides insights into the cycles of gas into and out of stars. Their period-luminosity relations also allow LPVs to act as distance measures.

### D.5. Beta-Lyrae

Beta-Lyrae ($\beta$-Lyrae) stars are close binary star systems in which the outer gaseous layers of both stars have expanded to the point that the pair is enveloped in a shared gaseous envelope. At this stage, gaseous material can be transferred from one star to another, altering either star's evolution. Their periods are typically a few days, and their light curves display continuous variations in brightness rather than flat maxima or minima like EA-type binaries, for example. Their dynamics provide direct information on gaseous mass transfer between binary stars, stellar structure under extreme tidal distortion, and the role of binarity in late stellar evolution.

### D.6. Blazhko

Blazhko variables are a sub-class of RR Lyrae variables, which exhibit the rare Blazhko effect in which their light curve amplitudes and phases are modulated over long time periods (tens to hundreds of days). In part owing to its rarity relative to normal RR Lyrae stars, there is not yet a consensus to the physical mechanism driving the Blazhko effect. The effect may be explained with magnetic fields or resonances within the star, so these stars are useful for studying exotic phenomena which can occur in stars.

### D.7. Type II Cepheids (Cepheid-II) and Anomalous Cepheids (ACEP)

Cepheid-II stars have old, low-mass stars with periods typically of tens of days and can produce a variety of light curve morphologies. They deviate from classical Cepheids because they are much fainter, but they do follow their own period-luminosity relation, enabling them to also be used as distance measures.

ACEPs have periods and luminosities inbetween those of RR Lyrae and classical Cepheids (roughly 0.3–2 days) with amplitudes of about 0.3–1.0 magnitudes, and their light curves typically resemble those of RRab stars. Their physical nature is not very well understood, but they have be proposed to be a product of gaseous mass transfer in a binary star system. ACEPs provide special astrophysical insights into stellar evolution pathways involving binary interaction, as well as into the environments where they typically occur.

### D.8. High-amplitude Delta-Scutis (HADS) and Low-amplitude Delta-Scutis (LADS)

Delta-Scuti ($\delta$-Scuti) stars are pulsating variables stars with short periods and are typically divided into the HADS and LADS subclasses based on the morphology of their light curves. HADS have simple, regular, sawtooth-like light curves with periods <0.3 days and amplitudes greater than ~0.3 magnitudes. In contrast, LADS exhibit complex, multi-periodic light curves with amplitudes below ~0.1 magnitudes. HADS provide clean tests of stellar pulsation theory and scaling relations, while LADS are testbeds for asteroseismology.

### D.9. Ellipsoidal Binaries (ELL)

ELLs are close binary star systems in which the stars are tidally distorted into ellipsoidal shapes, producing photometric variability without eclipses. ELL light curves have smooth, nearly sinusoidal variations with two unequal minima per cycle, arising from the elongated parts rotating in and out of view. They are astrophysically important because they reveal details of binary star evolution, stellar shapes, and the presence of companion objects such as white dwarfs, neutron stars, or black holes.

### D.10. Hump variables

The Hump class is used as a catch-all for the small amount of periodic variables which (Drake et al., 2014) were unable to classify into any other known classes but do show clear periodic variability. Some of these objects exhibit vaguely sawtooth-like variability, like what is seen in RRab stars, but others have smoother variability.

### D.11. Post-Common-Envelope Binaries (PCEB)

PCEBs are close binary stars that have recently emerged from a common-envelope evolutionary phase, in which one star expanded and engulfed its companion star inside its expanded outer layers ("envelope"). Their light curves show a wide range of morphologies, including eclipses and ellipsoidal modulations, depending on the physical properties of the system like the inclination of the orbit relative to our line-of-sight and the nature of either of the stars in the binary system. Their periods tend to be very short as the common-envelope phase drives angular momentum out of the orbit. The smaller star in these systems are often a white dwarf, so these systems are an opportunity to study potential progenitors of Type Ia supernovae.

### D.12. EA with unknown period (EAup)

EAup are EA-type stars where (Drake et al., 2014) were unable to determine their periods for any reason.

# E. Complete Set of Confusion Matrices for Classification

*Figure 13.* Confusion matrix of `Chronos` on four classifiers.

To provide detailed information about classification performance across different variable star classes, we report the confusion matrices for all embeddings and the hand-crafted features in this section. For the random forest and MLP classifiers, the per-class accuracies are obtained by averaging over the results of 10 runs.

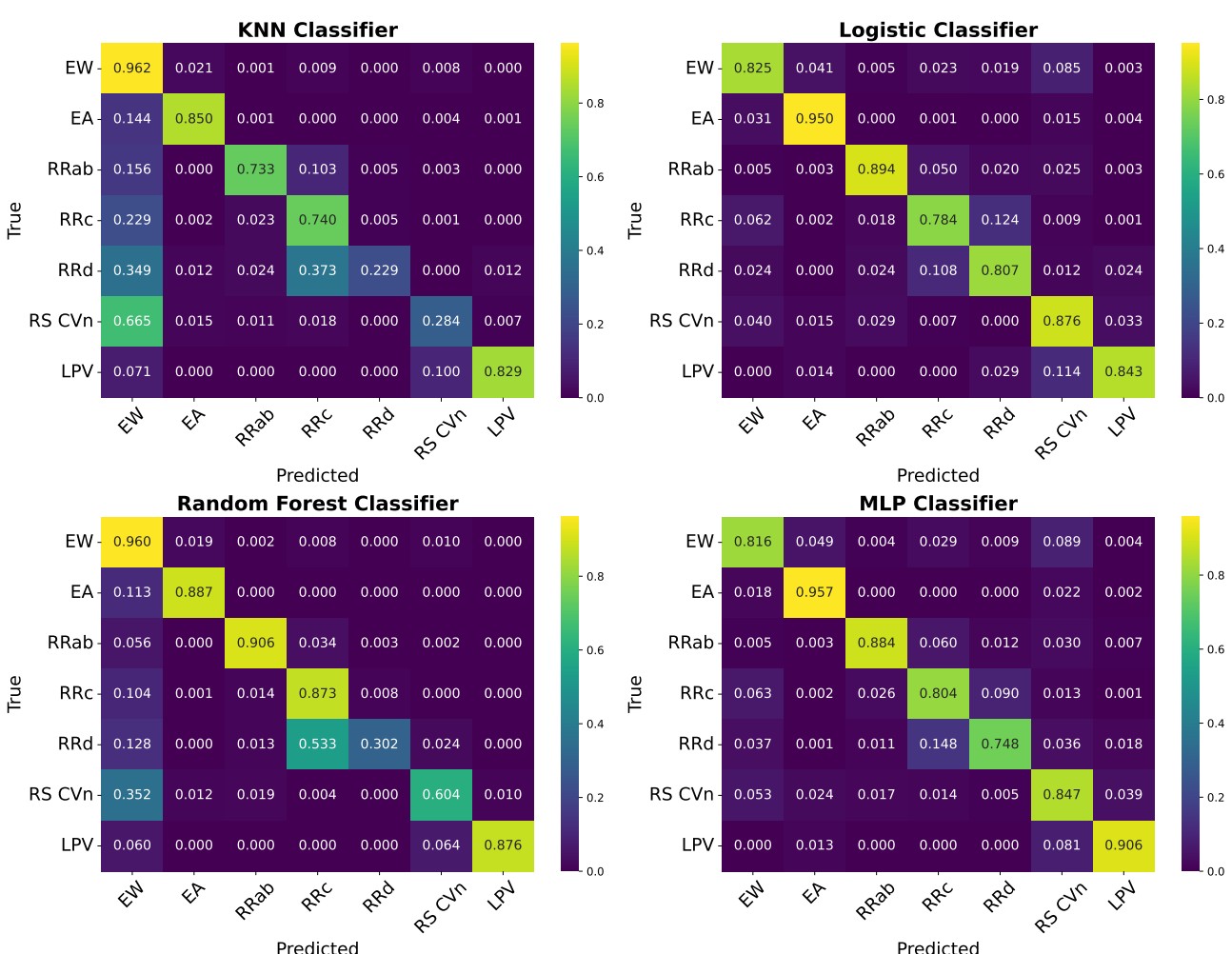

*Figure 14.* Confusion matrix of hand-crafted features on four classifiers.

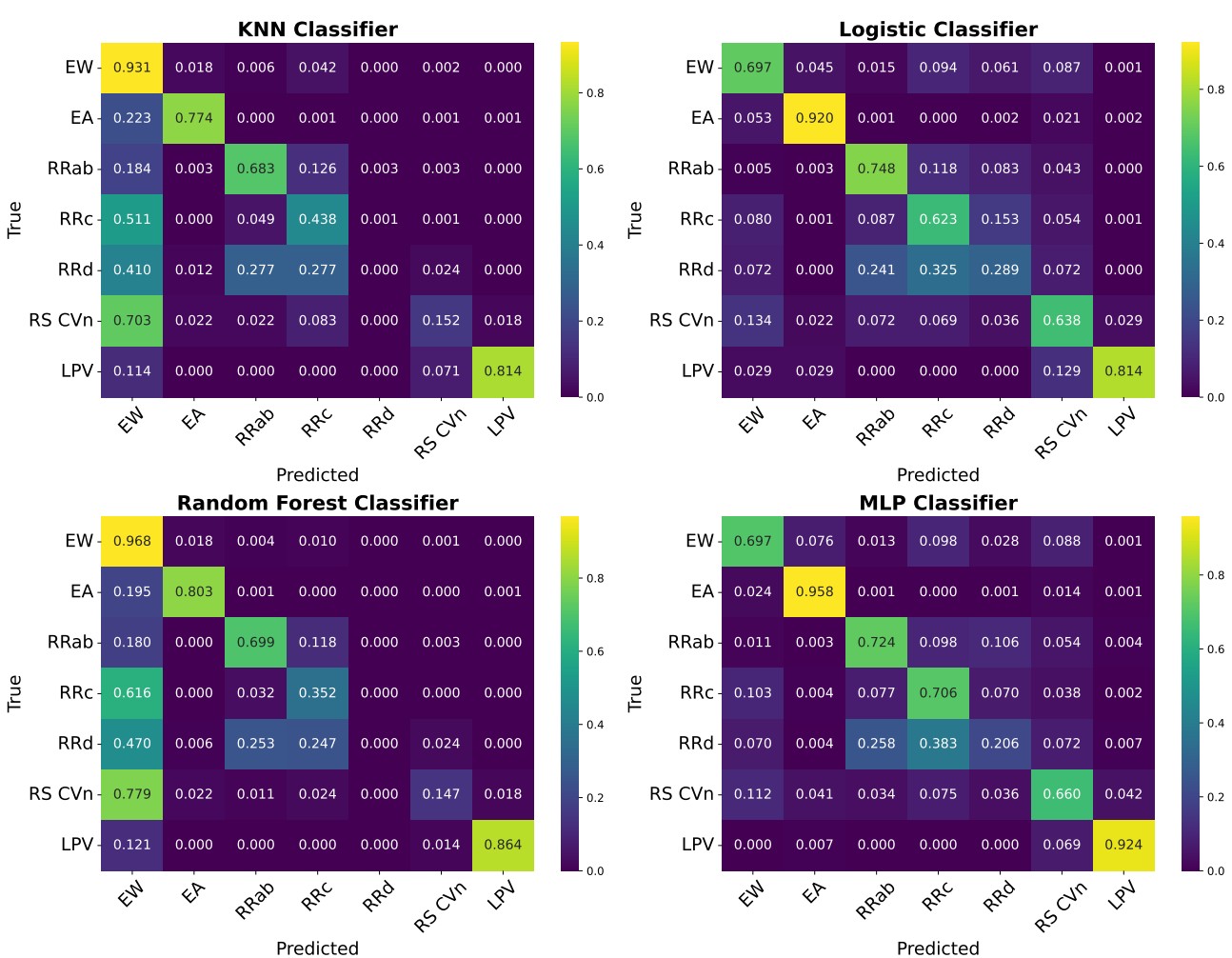

*Figure 15.* Confusion matrix of `Chronos-Bolt` on four classifiers.

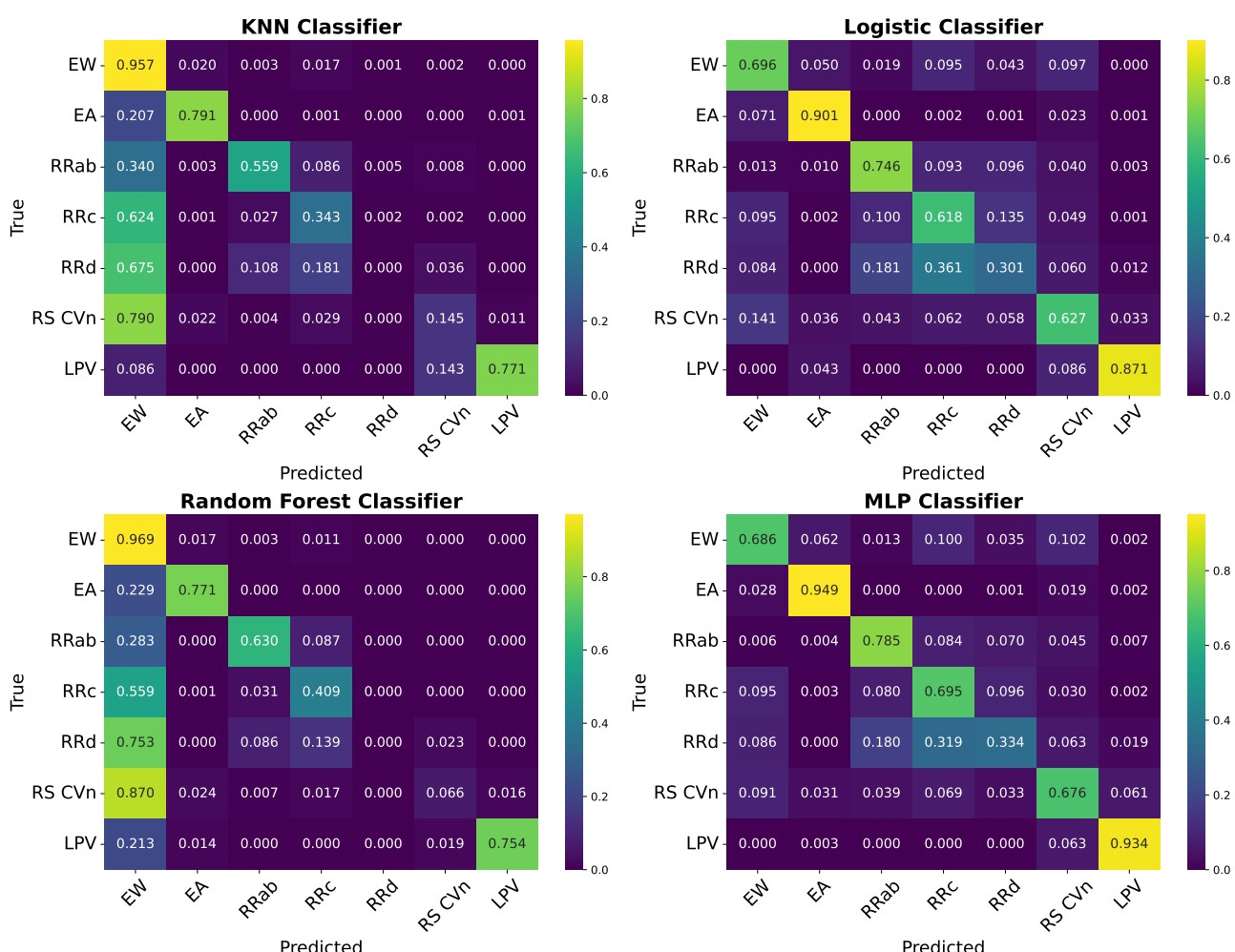

*Figure 16.* Confusion matrix of `Moirai` on four classifiers.

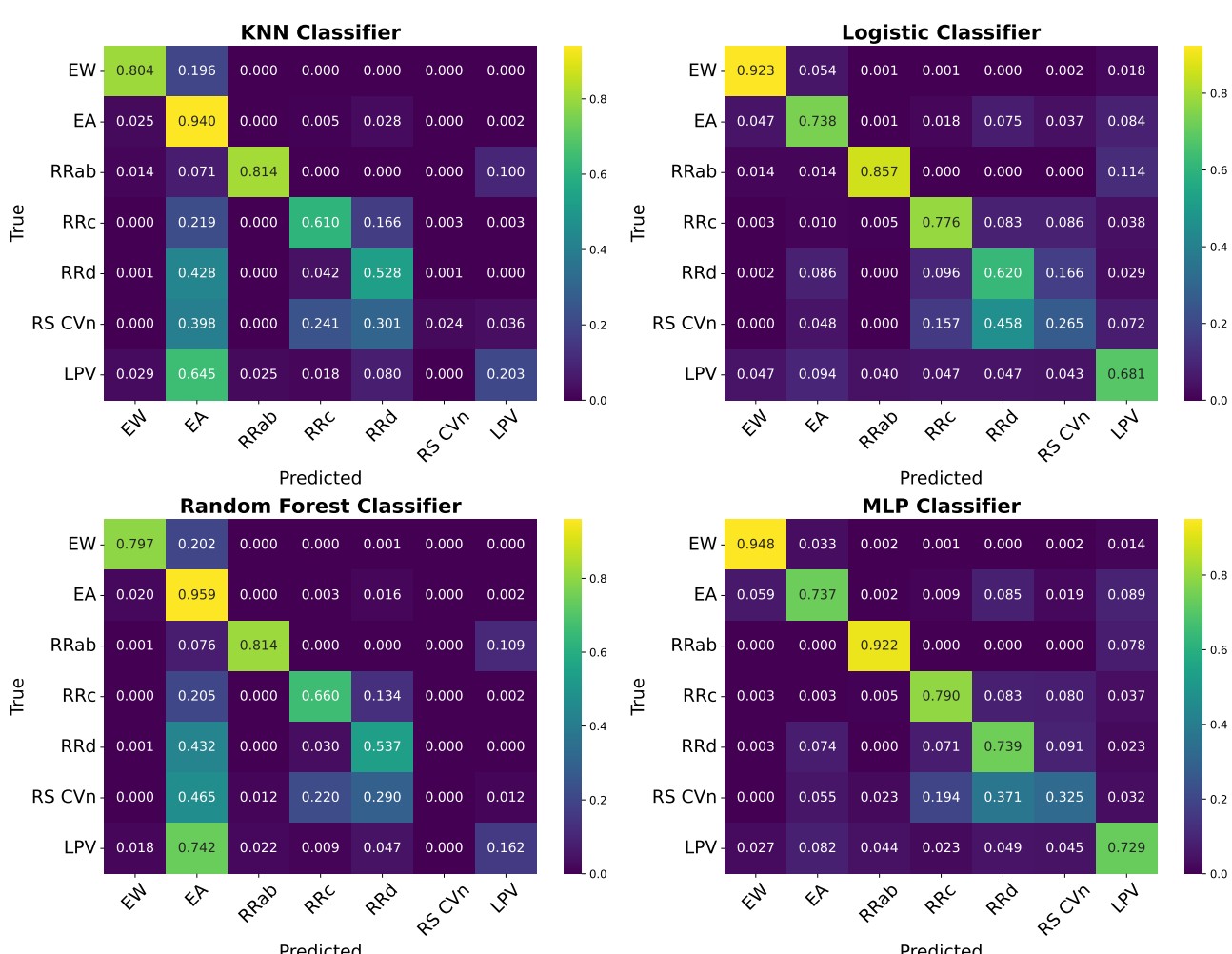

*Figure 17.* Confusion matrix of `Time-MoE` on four classifiers.

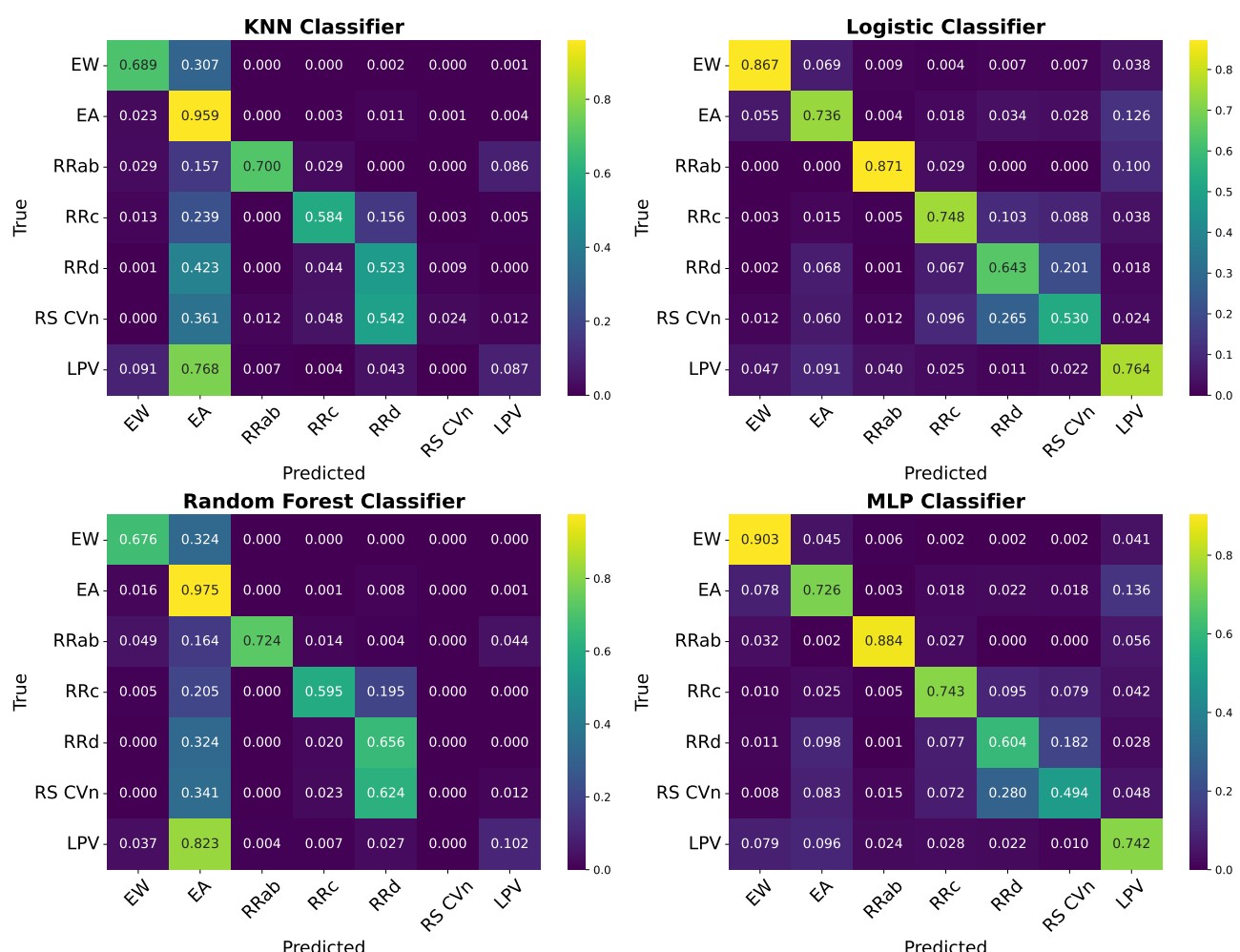

*Figure 18.* Confusion matrix of `Mantis+` on four classifiers.

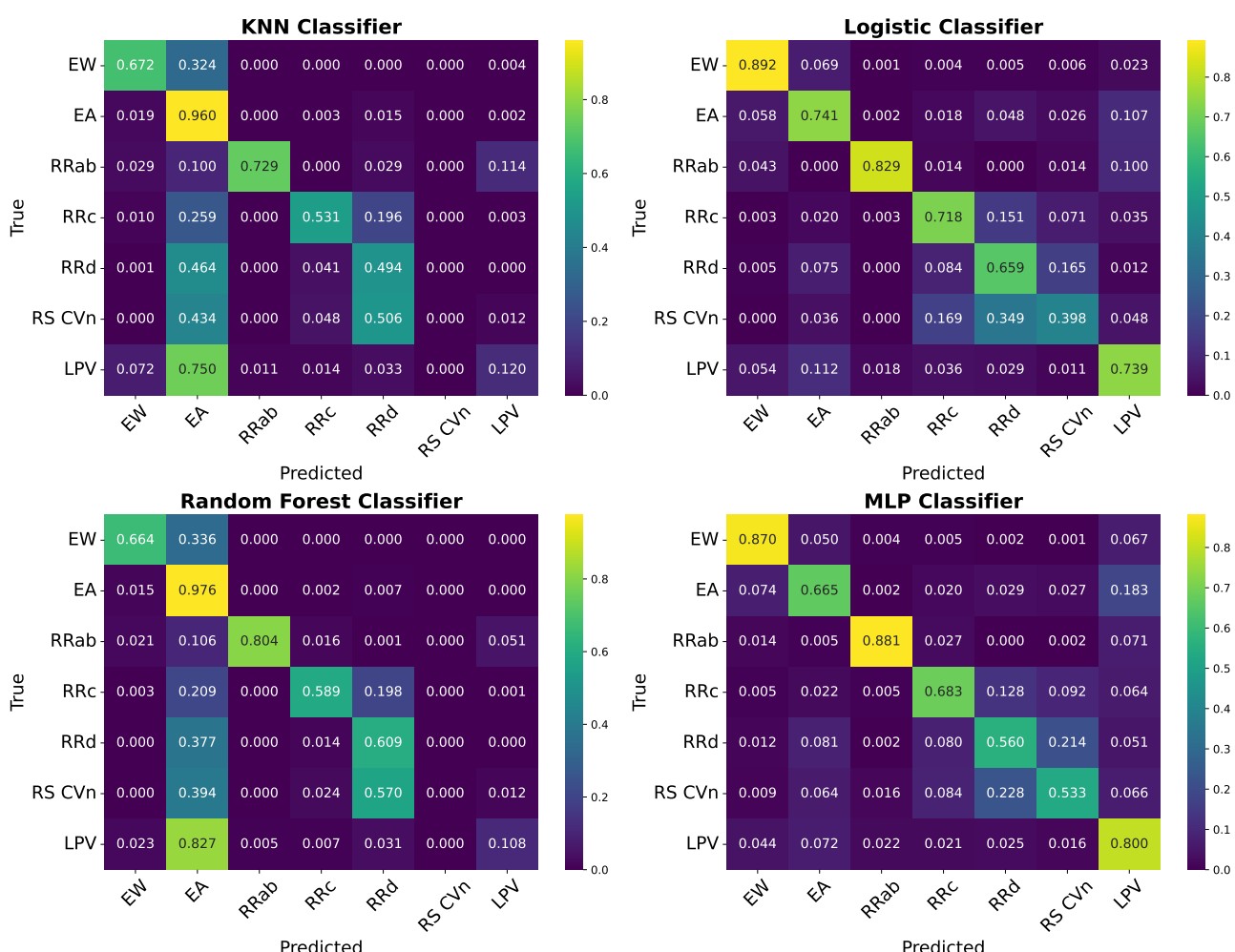

*Figure 19.* Confusion matrix of `MantisV2` on four classifiers.

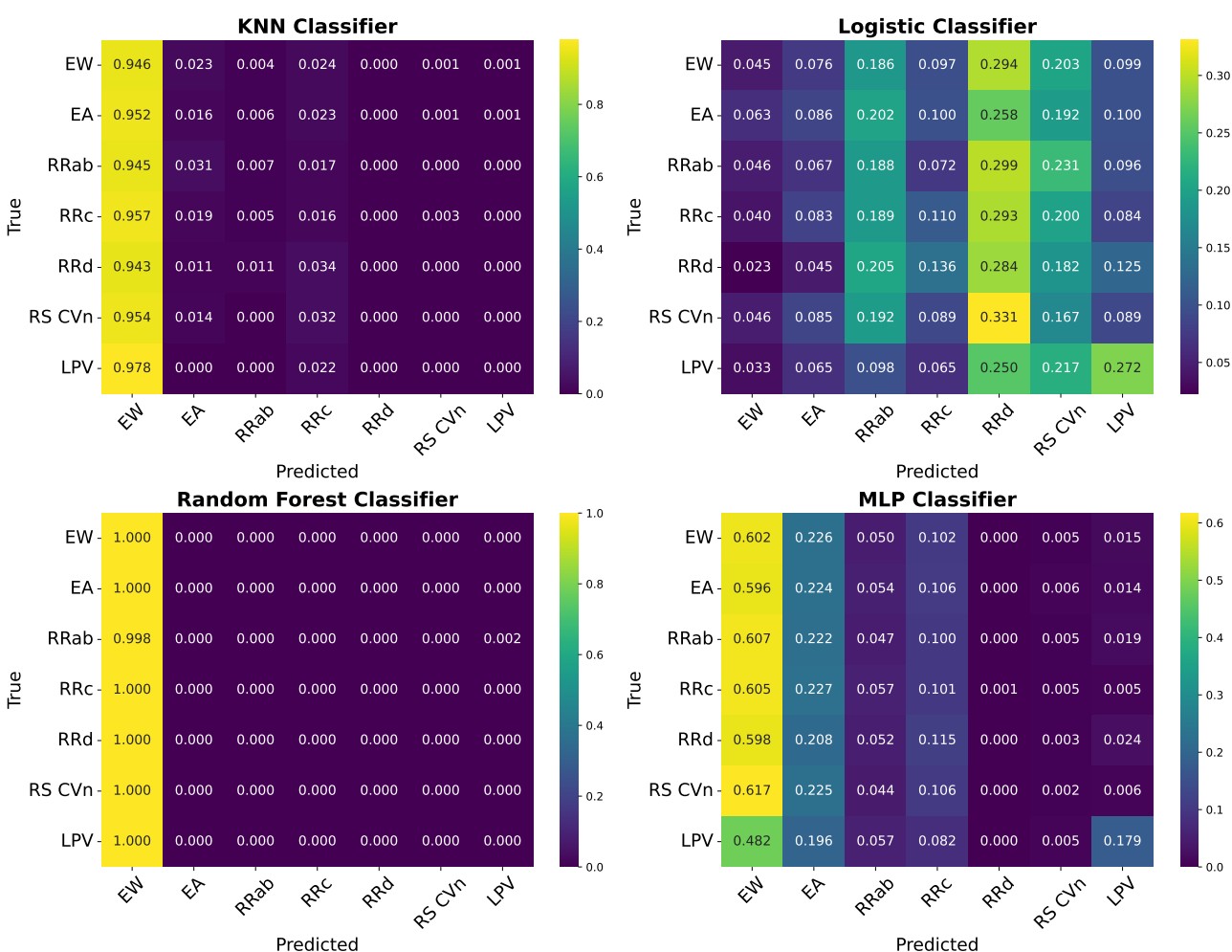

*Figure 20.* Confusion matrix of `Astromer-1` on four classifiers.

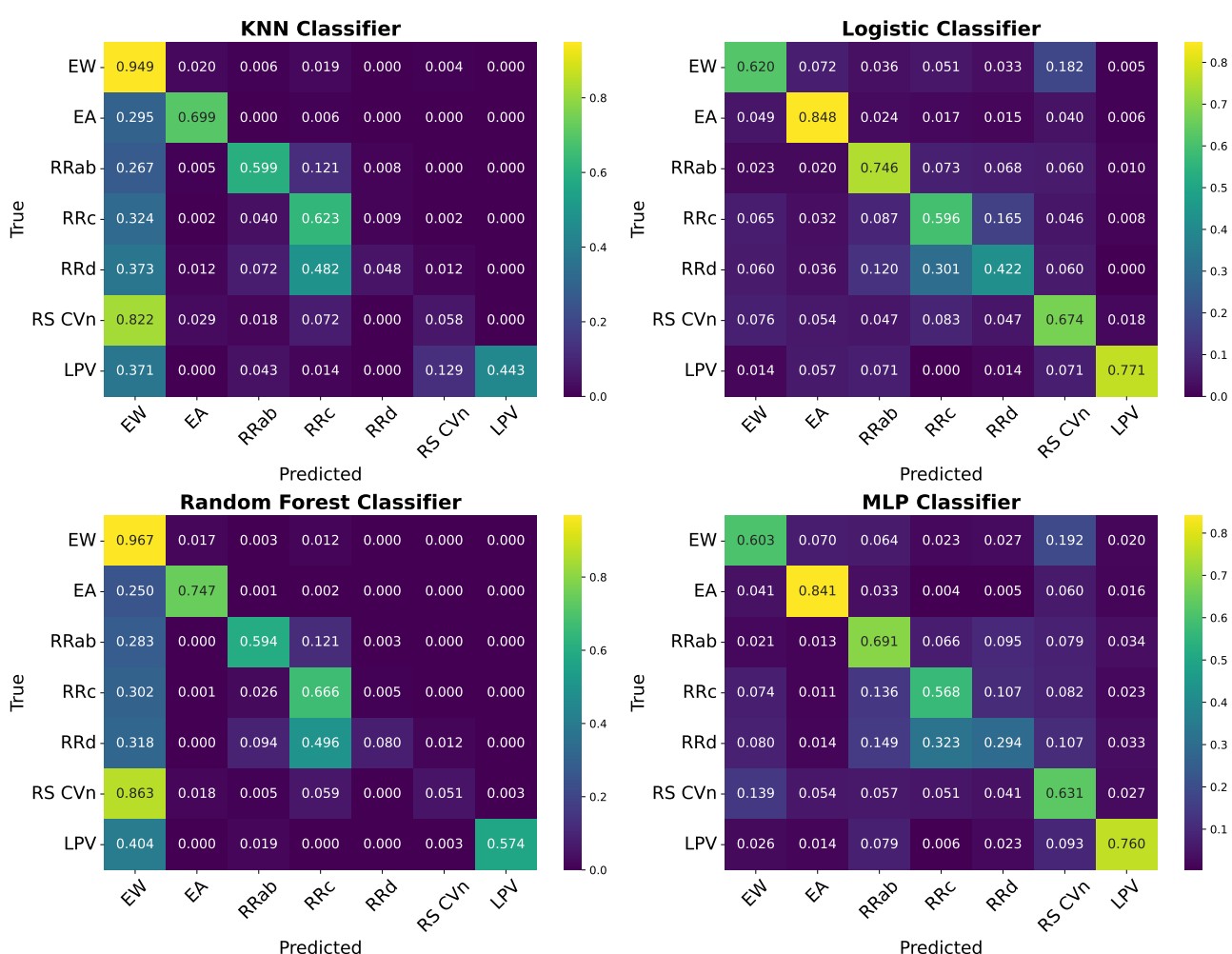

*Figure 21.* Confusion matrix of `Astromer-2` on four classifiers.

