# OpenReview forum: "StarEmbed: Benchmarking Time Series Foundation Models on Astronomical Observations of Variable Stars"
_ICML.cc/2026/Conference — ICML 2026 regular_

### Official Review · Reviewer_XsqF · 2026-03-09

**Soundness:** 3
**Presentation:** 3
**Significance:** 3
**Originality:** 3
**Overall Recommendation:** 4
**Confidence:** 2

**Summary:**

This paper introduces StarEmbed, presented as the first public benchmark for evaluating time series foundation models on astronomical light curves. The benchmark contains roughly 40,000 expert-labeled ZTF light curves across seven classes, with fixed train/validation/test splits, and evaluates embeddings on three downstream tasks: unsupervised clustering, supervised classification, and out-of-distribution source detection. The paper compares three TSFM families, including Moirai, Chronos, and Time-MoE and a domain-specific astronomy model Astromer and hand-crafted feature baselines widely used in astrophysics. The main finding is that Chronos is strong overall, matching hand-crafted features on clustering and outperforming them on OOD detection.

**Compliance With Llm Reviewing Policy:**

Affirmed.

**Final Justification:**

Authors solved most of my concerns and the novelty of this paper is high. I maintain that the weak-accept score of 4 is largely due to unfamiliarity with this area.

**Key Questions For Authors:**

1) Can the authors provide a more systematic robustness analysis of the benchmark itself, including sensitivity to dataset split, class imbalance handling, model size, embedding dimension, and the design choices?

2) Can the authors include ablations on key evaluation choices, such as preprocessing, class selection/filtering, downstream classifier setup, and OOD detector design, to clarify which conclusions are robust and which are benchmark-dependent?

3) Can the authors extract more actionable insights from the results, beyond reporting which model performs best? For instance, what properties of the data or embeddings appear to explain why some models perform better on clustering, while others are stronger on classification or OOD detection?

4) Can the authors clarify how reproducible the reported results are in practice? In particular, is there any analysis of run-to-run variance, environment dependence, or cross-system reproducibility that would help establish StarEmbed as a reliable long-term benchmark?

**Limitations:**

Yes, the paper does discuss limitations in substance, especially the mismatch between TSFM pretraining assumptions and the irregular, heteroskedastic nature of astronomical light curves.

**Strengths And Weaknesses:**

Strengths:

1) The strongest aspect of the paper is its dataset and benchmark contribution for the specific science domain. It is also very novel and timely work for the specific astronomical light curves data: irregular sampled, multi-band, noisy and large-scale, which may different from some regularly sampled time series data and demonstrate new features.

2) The downstream applications are broader including clustering, classification, and OOD. Three types of applications are involved in the experiments, and it shows that different models demonstrate differently among these tasks.

Weakness:

1) As a benchmark work, this work lacks the ablation test on the benchmark design,;no clear sensitivity to dataset split, class imbalance handling, model size, embedding dimension; no robustness analysis on the OOD task.

2) This work is primarily an evaluation paper rather than a methodological advance. If it can include results like how to further improve the task performance for each TSFM model, under which circumstance which model can be a better choice, give some strategy guidance would make it more meaningful and insight for a benchmark work.

3) A benchmark work is better to be stable enough for the future work to build on it. This work lacks such analysis, and we don’t know its performance on different system and reproducibility.

---

> ### Author Rebuttal · Authors · 2026-03-31
>
> We thank the reviewer for raising thought-provoking points that have allowed us to improve this paper. The reviewer raises three weaknesses (W1, W2, W3) and four questions (Q1, Q2, Q3, Q4), which we answer thematically below to address the robustness of our results, some actionable insights, and the reproducibility of what we have done.
>
> ## 1. Robustness analysis (**W1, Q1, Q2**)
> > “lacks ablation test on the benchmark design; no clear sensitivity to dataset split, class imbalance handling, model size, embedding dimension; no robustness analysis on the OOD task”
>
> * Our data splits are done independently by class to preserve the class imbalance inherent to the dataset (see **Sec. 3**). In **Fig. 2**, we show that the distribution of sampling frequencies and target measurement uncertainties across both variates are the same for train and test.
> * We initially used small TSFMs to maintain a "fair" comparison with domain-specific models (e.g., Astromer 2 at ~5M parameters). We have now added new classification results using all model sizes from the Chronos and Moirai families. Larger models/embedding sizes show generally similar classification performance and are still worse than the handcrafted features (see https://imgur.com/a/XdCSxFv)
> * We add a **new OOD evaluation method using the Local Outlier Factor (LOF)**. The LOF findings are broadly consistent with those from the IF. We additionally find that TSFM embeddings yield very strong OOD performance when paired with LOF, still well outperforming the handcrafted features. The **full result table** is in https://imgur.com/a/gcpRz2D.
>
> ---
>
> ## 2. Actionable insights (**W2, Q3**)
> > “how to further improve the task performance for each TSFM model, under which circumstance which model can be a better choice, give some strategy guidance”
>
> > “Extract more actionable insights; what properties explain [the performance across tasks]?”
>
> + **Fig. 4** illustrates how TSFMs not treating irregular sampling properly prevents them from being able to infer the period (**Appx. H**), the most important feature determining the handcrafted features' performance (**Appx. F**). In **Sec. 6**, we comment on how a TSFM with a better treatment of irregular sampling should excel in StarEmbed. We add a **new validation of two techniques** for doing so on a toy sinusoidal dataset: (i) **Rotary Positional Encoding (RoPE)** on the irregular observation timestamps (instead of the regular token/patch ID), and (ii) using the **measurement $\Delta t$ as a covariate**. The results show that our RoPE + true timestamp implementation ($R^2 = 0.188$) outperforms $\Delta t$ as a covariate ($R^2 = 0.044$) and not treating irregular sampling (vanilla; $R^2 = 0.016$). See the **full setting and forecasting plot** at https://imgur.com/a/YPfHmTw. These findings demonstrate that alternative techniques for encoding time, different from those currently used by TSFMs, could improve benchmark performance with future model development.
> + In another new experiment, we visualize the **attention patterns** of Chronos and Moirai (https://imgur.com/a/0Rnfdlf). These figures suggest that TSFMs with point-wise tokenization perform better than TSFMs with patch-wise tokenization because the point-wise models better attend to the brightness extrema of the time series, capturing the full range of variability. The patch-based models group neighboring measurements, which may exaggerate the irregular sampling challenge, making it difficult to learn from such heterogeneous patches.
>
> ---
>
> ## 3. Reproducibility (**W3, Q4**)
> > “how reproducible are the reported results? Is there any analysis of run-to-run variance, environment dependence, or cross-system reproducibility”
>
> + All non-deterministic evaluations are run with 10 random seeds. Standard deviations for metrics are reported in **Tables 2, 3, and 4**.
> + The code base to be made public will include detailed environment setup and dependency installation instructions, as well as READMEs to guide an external user through using the code.
> + During development, we have verified that scripts run on different Python versions, building confidence for cross-system reproducibility.
>
> ---
>
> Thanks again for your valuable feedback and insights. Please do not hesitate to let us know if there are any other aspects that you would like us to clarify.

---

> > ### Author Rebuttal · Reviewer_XsqF · 2026-04-01
> >
> > Thanks for the explanation from authors. For robustness and reproducibility, my concerns have been all addressed. For the actionable insights, however, I am still not seeing a clear answer to "under which circumstance which model can be a better choice, give some strategy guidance". It maybe that this question is beyond the scope of this paper and need more time to address. I will therefore maintain my score of 4 weak accept, mainly because I still lack of confidence in this area. Thanks for understanding.

---

> > > ### Author Response · Authors · 2026-04-03
> > >
> > > We are glad to hear that our rebuttal has addressed the concerns on robustness and reproducibility. We understand your remaining concern regarding the actionable insights resulting from our work (i.e. "under which circumstance which model can be a better choice, give some strategy guidance"). Please allow us to clarify here:
> > >
> > > Our results from rebuttal experiments and the original text support 5 pieces of concrete guidance for model selection and future TSFM development in the context of irregularly sampled astronomical time series:
> > > 1. Hand-crafted features are the top performer in the clustering and classification tasks; they are a good choice when performance is the highest priority. Chronos(-Bolt) with Local Outlier Factor (see **W1 response to [bjHB](https://openreview.net/forum?id=Ujf6Is4cdt&noteId=Djn2RjmYeV)**) are the best choice for OOD detection or when computational cost is a concern. Our new runtime analysis (see **Q4 response to [bjHB](https://openreview.net/forum?id=Ujf6Is4cdt&noteId=Djn2RjmYeV)**) shows that hand-crafted features are inviable at scale. Other existing TSFMs do not have use cases over these options; the remainder of our guidance relates to future TSFMs development.
> > > 2. Point-wise tokenization generally outperforms patch-wise tokenization. Our new attention tracking experiments (see **Q1 response to [kug1](https://openreview.net/forum?id=Ujf6Is4cdt&noteId=bVsqLCjHkt)**) suggest that patch-wise tokenization appears to be more vulerable to variable sampling frequencies which make the patches very heterogeneous while point-wise tokenization is more robust to this. Our classification and clustering results are also consistent with this finding.
> > > 3. Existing positional encoding schemes (RoPE) do not work very well out-of-the-box on irregularly sampled time series, but intelligent and targeted modifications yield considerable performance gains (new experiment results in **[2. Actionable Insights response](https://openreview.net/forum?id=Ujf6Is4cdt&noteId=9PeUpbsWe3)**)
> > > 4. Existing TSFMs for classification (e.g., Mantis) do not necessarily give better results (see **W1 response to [kug1](https://openreview.net/forum?id=Ujf6Is4cdt&noteId=bVsqLCjHkt)**).
> > > 5. Fine-tuning with a classification task yields better downstream-task performance than fine-tuning with a forecasting task (see **W2 response to [knTX](https://openreview.net/forum?id=Ujf6Is4cdt&noteId=pxOXmRHLQ0)**).
> > >
> > > These findings are pioneering pieces of guidance for the development and usage of TSFMs on irregularly sampled astronomical time series.
> > >
> > > ---
> > >
> > > Thank you again for your time and feedbacks. We hope these clarifications solve your concerns.

---

### Official Review · Reviewer_kug1 · 2026-03-11

**Soundness:** 3
**Presentation:** 3
**Significance:** 3
**Originality:** 2
**Overall Recommendation:** 4
**Confidence:** 3

**Summary:**

This paper investigates the application and performance of time series foundation models within the field of astronomy. To facilitate this study, the authors introduce StarEmbed, a novel benchmark dataset comprising around 40,000 expertly labeled light curves spanning seven distinct astrophysical classes. Leveraging this dataset, the authors conduct a comprehensive evaluation encompassing unsupervised clustering, supervised classification, and out-of-distribution (OOD) source detection. The findings reveal that despite not being pre-trained on astronomical observational data, TSFMs can outperform existing domain-specific models in certain tasks. Notably, TSFMs, particularly those in the Chronos family, demonstrate exceptional proficiency in clustering and OOD detection, significantly surpassing specialized models like Astromer. While TSFMs do not yet consistently exceed hand-crafted feature-based methods in supervised classification, their strong generalization capabilities underscore their potential.

**Compliance With Llm Reviewing Policy:**

Affirmed.

**Key Questions For Authors:**

In scientific research, interpretability is often as important as accuracy. Given the inherent black-box nature of TSFMs, did the authors conduct any exploration into their interpretability?

**Limitations:**

This paper presents a scientific benchmark dataset and the authors did not identify any potential negative societal or ethical issues.

**Strengths And Weaknesses:**

- **Strengths**
    1. The paper is well written and easy to follow.
    2. This paper makes a significant contribution by introducing StarEmbed, a vital new dataset for time series research. Through a series of diverse experiments, it yields several noteworthy results that highlight the potential of foundation models in astronomy.
    3. The paper highlights an interesting finding: despite not being specifically designed to handle irregularly sampled data, TSFMs are still able to deliver competitive results. This not only demonstrates the robustness of these models but also points to potential areas for future improvement.

- **Weaknesses**
    1. Time series foundation models such as Chronos are primarily designed for forecasting tasks, without specific consideration for pattern recognition tasks like clustering and classification. The paper did not test foundation models specifically designed for time series classification, which might further improve accuracy on these tasks.

---

> ### Author Rebuttal · Authors · 2026-03-31
>
> We thank the reviewer for their insights, which have allowed us to make important modifications to the paper. The reviewer raises one weakness (W1) and one key question (Q1).
>
> ---
>
> ## Responses to weaknesses
> > **W1:** The paper did not test TSFMs specifically designed for time series classification.
>
> We have **added benchmarking of two SOTA classification-specific TSFMs** (MantisV2, Mantis+ [1,2]), including both zero-shot and fine-tuning results for the supervised classification task. The table below shows the results alongside classification performance for Chronos and the handcrafted features. Each metric is averaged separately over 10 runs. Detailed tables can be found here: zero-shot classification comparison (https://imgur.com/a/RRRTkC6), clustering comparison (https://imgur.com/6Iy5FYy), and zero-shot vs. fine-tuned Mantis comparison (https://imgur.com/a/1GItXfZ).
>
> | Model                  | Setting                              | Accuracy        | Precision       | Recall          | F1              |
> |--|--|---:|---:|--:|--:|
> | Chronos                | Zero-shot                            | 0.783   | 0.589   | 0.758    | 0.643   |
> | Chronos-FTcls          | Fine-tuned for classification        | *0.778*  | 0.613   | 0.766  | *0.659*   |
> | MantisV2               | Zero-shot                            | 0.686   | 0.551    | 0.714   | 0.570 |
> | MantisV2-FT            | Full fine-tuning + MLP head          | 0.773   | 0.605   | 0.780   | 0.638  |
> | Mantis+                | Zero-shot                            | 0.731   | 0.568 | 0.728  | 0.600  |
> | Mantis+-FT             | Full fine-tuning + MLP head          | 0.770   | *0.619*   | *0.790*   | 0.646  |
> | Hand-crafted features  | Benchmark baseline                   | **0.833**  | **0.672**  | **0.851**  | **0.723**  |
>
>
> **In conclusion**, we find that all Mantis models performed worse than handcrafted features and Chronos in the zero-shot setting. Fine-tuning brings improvements to Mantis with the MLP head, yet it is only roughly on par with Chronos-level performance rather than clearly surpassing it.
>
> Beyond classification-specific TSFMs, we also explore fine-tuning Chronos and making architecture refinements to improve the treatment of irregular sampling in these TSFMs. See **response W2 to Reviewer knTX** and https://imgur.com/a/7kiMGXY for Chronos fine-tuning results. See the "Actionable Insights" response to **Reviewer XsqF** and https://imgur.com/a/YPfHmTw for architecture modifications.
>
> [1] "Mantis: Lightweight Calibrated Foundation Model for User-Friendly Time Series Classification.", ICML FMSD Workshop (2025)
>
> [2] "MantisV2: Closing the Zero-Shot Gap in Time Series Classification with Synthetic Data and Test-Time Strategies", ICLR TSALM Workshop (2026)
>
> ---
>
> ## Responses to questions
> > **Q1:** Did the authors conduct any exploration into [TSFMs'] interpretability?
>
> The original manuscript included basic, sometimes qualitative, investigations into the interpretability of the various embedding models (**Appx. B, E, F, H**).
>
> In short, we hypothesize that the primary shortcoming of the TSFMs is their inability to treat the irregular sampling of the time series. **Fig. 4** shows how this destroys the morphology of the periodicity. **Appx. F** demonstrates that the most important hand-crafted feature is the period, so the inability of TSFMs to capture this imposes a critical ceiling on their performance. The period regression tests in **Appx H** confirm this; TSFM embeddings cannot be used to infer the period. Discussion of this hypothesis and these experiments is present in **Sec. 6**.
>
> We have also quantitatively extended this analysis by **adding a new interpretability test** of the TSFMs using their attention scores.
>
> We visualize the attention scores of Chronos and Moirai to show the difference in their abilities. In these **Figures** (https://imgur.com/a/0Rnfdlf), we see that Chronos (which inputs point-wise measurements) concentrates attention on brightness extrema, capturing the full range of variability, including quick/slow changes in brightness. Moirai (a patch-based model) groups neighboring measurements, which will exacerbate the challenges associated with irregular sampling, as each patch contains an extremely different time baseline, making it challenging to learn the underlying structure within the data.
>
> ---
>
> Thank you again for the valuable feedback. We hope our clarification addresses your concerns and welcome further discussion!

---

> > ### Author Rebuttal · Reviewer_kug1 · 2026-04-05
> >
> > Thank you for the author's response. The rebuttal regarding the interpretability part addressed my concern.

---

> > > ### Author Response · Authors · 2026-04-07
> > >
> > > Thank you for your feedback. We are glad that our response has resolved your concerns regarding interpretability. In the first round of rebuttal, we addressed your concerns about TSFMs designed for classification. Specifically, we:
> > > - added results on SOTA TSFMs designed for classification (MantisV2 and Mantis+) **(Response to W1)**.
> > >
> > > Could you please let us know whether our rebuttal has also resolved your concerns about TSFMs designed for classification?
> > >
> > > If there are any remaining concerns, we would be very happy to discuss them further. As a gentle reminder, you may include additional comments by using the edit function in the Acknowledgement box, and we can address them by further revising this response.
> > >
> > > ---
> > >
> > > Thank you again for the time and effort you have devoted to reviewing our paper.

---

### Official Review · Reviewer_bjHB · 2026-03-11

**Soundness:** 4
**Presentation:** 3
**Significance:** 4
**Originality:** 4
**Overall Recommendation:** 5
**Confidence:** 3

**Summary:**

This paper introduces StarEmbed, a time series benchmark for evaluating time series foundation model embeddings on classification, clustering, and OOD detection tasks on astronomical time series. The benchmark is created by filtering ZTF observations with human-labeled stars from the CSPVS catalog. Simple baselines, domain specific TSFMs, and general TSFMs are evaluated on the benchmark on the three tasks. with various classification methods trained on top of the embeddings. The authors find that while expensive hand crafted features perform best, general TSFMs work quite well and strongly outperform domain specific TSFMs Astromer-1 and 2.

**Compliance With Llm Reviewing Policy:**

Affirmed.

**Final Justification:**

The rebuttal has addressed my concerns due to inclusion of further finetuning results, more clustering/detector approaches, and further hypotheses to support future work in this area. The benchmark on its own is a solid contribution, and therefore I raise my score by one point.

**Key Questions For Authors:**

1.  On the supervised classification task for example, how would TSFMs perform after finetuning when fitted with a classification head, rather than using continual pretraining or training on their embeddings? This seems like a reasonable baseline that might be done with TSFMs, and would be helpful in understanding for others on where to begin with finetuning work or decoupling the classification approach from the model (W1, W2)
3. How do generic LLMs or VLM embeddings perform on StarEmbed tasks? This would help understand the relative strength of the TSFMs on this task, and strengthen the claim that TSFM embeddings are truly different. (W4)
3. Are there any hypotheses as to why Chronos outperforms all other TSFMs? The general TSFMs (and Astromer-2) have a relatively small difference in scores, which makes me wonder what the differences in their predictions may be due to. (W3)
4. Is there any way to measure the relative cost of creating hand-crafted features, and why we would not want to use it generally over TSFMs?

**Limitations:**

Yes

**Strengths And Weaknesses:**

Strengths
- StarEmbed presents a new specialized benchmark with ~40k labeled stars for stellar light curves, which is important for the AI+Science community.
- Evaluates over a wide range of classification heads and model types.
- Time series within StarEmbed are highly irregular and heteroskedastic, which pose new challenges for TSFMs.
- Paper is generally well-written and easy to read.

Weaknesses
- Though a natural consequence of testing embedding quality, many of the results are closely tied to specific clustering or detector approaches, e.g. OOD detection is closely tied to isolation forest, making it difficult to fully attribute performance to TSFMs alone.
- Finetuning seems like a natural fit for this benchmark, but only a limited simple experiment is done with Moriai.
- While this work focuses on TSFM embeddings, it does not test established methods for time series classification [1], such as with VLMs or LLMs.
- There are few case studies to interpret where errors occur among current general or domain-specific TSFMs, limiting understanding of their performance.

[1] Daswani, M., Bellaiche, M. M., Wilson, M., Ivanov, D., Papkov, M., Schnider, E., ... & Telang, U. (2024). Plots unlock time-series understanding in multimodal models.

---

> ### Author Rebuttal · Authors · 2026-03-31
>
> We thank the reviewer for questions that have helped us improve this paper. The reviewer raises four weaknesses (W1, W2, W3, W4) and four questions (Q1, Q2, Q3, Q4).
>
> ## Responses to weaknesses and questions
>
> > **W1:** “...many of the results are closely tied to specific clustering or detector approaches... OOD detection is closely tied to isolation forest...”
>
> We agree other methods should be included, and added a table with results from a **new Local Outlier Factor (LOF) test**. We choose LOF because it imposes a different inductive bias than isolation forest (IF). IF is tree/partition-based, while LOF is density-based and tests a different aspect of the embedding geometry. The full table (https://imgur.com/a/gcpRz2D) confirms that, while the absolute purity values differ across IF and LOF, the **relative trend is consistent**. We can now show that the OOD results are robust to the OOD algorithm.
>
> In the other tasks, our embedding evaluation is not tied to a single downstream method: we use two clustering algorithms and four classifier families spanning non-parametric, linear, tree-based, and non-linear models. This provides a broad and robust evaluation of embedding quality.
>
> We hope the additional tests address the concern, but please let us know if another specific test would make the results even more robust.
>
> ---
> > **W2**: Finetuning seems like a natural fit for this benchmark...
> > **Q1**:  ...how would TSFMs perform after finetuning when fitted with a classification head...
>
>
> We added additional fine-tuning experiments on Chronos. Please see the **response to W2 from Reviewer knTX**.
>
> ---
> > **W3**. ...not test established methods for time series classification [1], such as with VLMs or LLMs.
> > **Q2**. How do generic LLMs or VLM embeddings perform on StarEmbed tasks?
>
> We respond to this weakness and question via two related points.
>
> * **TSFMs for classification.** We agree that more recent TSFMs for classification may provide improvements over Chronos and Morai and have added SOTA classification-specific TSFMs in both zero-shot and fine-tune settings, as noted in our **response to W1, Reviewer kug1**.
>
> * **LLM/VLMs.** LLM/VLMs are not a natural baseline for this benchmark. First, they include models that have >10^4 more parameters than TSFMs, making it infeasible to scale inference on astrophysical surveys with >10^7 light curves.
>
>     Second, they introduce additional modality conversions, extra design choices, prompt engineering, and preprocessing assumptions that are orthogonal to the question we study.
>
>     Third, recent studies show that LLMs struggle with time-series analysis [1, 2]. GPT-3/LLaMA-2 (6.7B - 175B) are worse than Chronos and Moirai (21M - 300M) in the Chronos paper (Sec. 5.5.2). For VLMs, the paper referenced [3] lacks a systematic comparison to well-established TSFMs on time series benchmarks.
>
>     Thus, we view LLM/VLM methods as an interesting future direction that is not yet the appropriate baseline for this benchmark.
>
> [1] Language Models Still Struggle to Zero-shot Reason about Time Series, arXiv:2404.11757
>
> [2] Are Language Models Actually Useful for Time Series Forecasting?, arXiv:2406.16964
>
> [3] Plots Unlock Time-Series Understanding in Multimodal Models, arXiv:2410.02637
>
> ---
>
> > **W4:** There are few case studies to interpret where errors occur among current general or domain-specific TSFMs
>
> To clarify, this paper includes several such case studies, see the **response to Reviewer kug1, Q2**. We also include new empirical analysis on the performance gap of Chronos and Moirai, see **Q3** (below).
>
> ---
>
> > **Q3:** Are there any hypotheses as to why Chronos outperforms all other TSFMs?
>
>
> We hypothesize the difference is related to how models tokenize the time series. Moirai uses patch-wise tokenization, whereas Chronos uses point-wise tokenization. Point-wise tokenization is less destructive when the raw sequence contains irregular gaps and nonuniform local (periodic) structure, because patching conflates consecutive observations separated by irregular time gaps.
>
> Our analysis of the attention pattern in Chronos and Moirai further verifies this hypothesis, see the **response to Reviewer kug1, Q1**.
>
> ---
>
> > **Q4:** “Measure relative cost of hand-crafted features (HCFs), and why we would not want to use it over TSFMs?”
>
> Handcrafted features are not a viable basis for the petabyte-scale era of variable star astronomy. Extracting the 138 features (most importantly, the costly Lomb-Scargle periodogram fit) takes >100 CPU core-hours for the ~40,000 stars in our dataset. Even with parallelization, this does not reasonably scale to >10^7 light curves provided by modern observatories. In contrast, with one A100 GPU, Chronos-tiny can embed our entire dataset in just 455 seconds. There are some techniques that can accelerate the feature extraction process, but many of the features already use highly performant Rust-based backends.
>
> ---
>
> Thanks for your time and thoughtful review!

---

> > ### Author Rebuttal · Reviewer_bjHB · 2026-04-03
> >
> > Thank you for the detailed rebuttal. My concerns have been addressed and I will raise my score by 1.

---

> > > ### Author Response · Authors · 2026-04-05
> > >
> > > Thank you for your consideration. We are glad that our rebuttal has addressed your concerns. We will carefully incorporate your comments and the new results into the paper. Your constructive feedback has been invaluable in strengthening this work.

---

### Official Review · Reviewer_knTX · 2026-03-13

**Soundness:** 2
**Presentation:** 3
**Significance:** 2
**Originality:** 3
**Overall Recommendation:** 3
**Confidence:** 5

**Summary:**

This paper introduces StarEmbed, a benchmark for evaluating time series foundation models (TSFMs) on astronomical light curves of variable stars. It compares several general-purpose TSFMs, including the Chronos family, Moirai, and Time-MoE, as well as a domain-specific transformer (Astromer) and traditional hand-crafted features. The benchmark is built from approximately 40,000 expert-labeled multi-band ZTF light curves spanning seven astrophysical classes. The evaluation covers three representative downstream tasks: unsupervised clustering, supervised classification, and out-of-distribution (OOD) detection. The main empirical finding is that the Chronos family performs strongly in this out-of-domain scientific time-series setting, achieving very competitive or leading results in clustering and OOD detection, although hand-crafted features remain the strongest approach for supervised classification.

**Compliance With Llm Reviewing Policy:**

Affirmed.

**Final Justification:**

I appreciate the effort the authors made and suggest further improvement of the original paper based on the substantial experiments added in the discussion phase.

**Key Questions For Authors:**

1. The paper deliberately focuses on frozen, zero-shot evaluation of TSFMs, which is a reasonable and clearly motivated choice. However, since Chronos is the strongest general-purpose model in several of the main experiments, could the authors clarify whether they attempted any form of Chronos adaptation (e.g., linear probing beyond the current setup, partial fine-tuning, or PEFT-style tuning), and if not, why such experiments were excluded? The appendix reports only a preliminary fine-tuning experiment on Moirai.
2. The paper refers to using “small” or “tiny” variants of Moirai, Chronos, and Time-MoE, but does not clearly summarize their parameter counts, architectural scale, or computational cost in one place. Could the authors add a concise table describing the exact variants used?
3. Using only small/tiny variants may influence the conclusions, especially since larger models might perform better and narrow the gap to hand-crafted features. Limiting model size for computational cost is reasonable, but the paper should more clearly justify this choice and report the exact scale of each variant.

**Limitations:**

yes

**Strengths And Weaknesses:**

Strengths
1. The evaluation setup is relatively comprehensive, covering three representative downstream tasks—clustering, classification, and OOD detection—thus providing a more complete assessment of representation quality than classification alone. At the same time, the authors present the results in a fairly restrained manner: rather than overstating the strengths of TSFMs, they explicitly acknowledge that hand-crafted features remain the strongest approach for supervised classification.
2. The paper is generally well written and clearly organized. The related work section provides a relatively systematic review of prior studies on variable-star analysis and TSFMs, and the paper positions itself appropriately as a benchmark/evaluation paper rather than a method paper centered on algorithmic novelty.
3. The research question is meaningful. The paper asks whether general-purpose TSFMs can transfer to astronomical variable-star light curves, which are characterized by irregular sampling, missing values, and heteroskedastic noise. This question is relevant to both scientific machine learning and time-series representation learning.
4. StarEmbed itself is a valuable benchmark resource. By curating and releasing roughly 40k expert-labeled ZTF light curves together with standardized splits and supporting resources, the paper helps promote more standardized evaluation and more reproducible research in this area.

Weaknesses
1. The paper provides fairly convincing evidence that general-purpose TSFMs are promising for clustering and OOD detection. However, the best results on supervised classification still come from hand-crafted features, which suggests that some of the stronger forward-looking claims may go slightly beyond what the current experiments directly support. For example, the results support competitiveness and strong zero-shot potential, but they do not yet fully justify broader implications such as replacing existing domain-specific pipelines or establishing clear superiority in the most central supervised setting.
2. Although the zero-shot / frozen evaluation setup is a reasonable and well-motivated design choice, the adaptation experiments in the appendix only cover Moirai and are limited to a relatively preliminary form of straightforward fine-tuning. For Chronos, which is the strongest general-purpose model in the main experiments, no fine-tuning or PEFT results are provided.
3. The current benchmark focuses on seven classes of periodic variable stars from ZTF, and is therefore better viewed as a valuable but relatively specialized benchmark. It remains to be validated whether the conclusions generalize to other surveys, non-periodic transients, or broader astronomical time-series tasks.
4. The paper uses 'small'/'tiny' model variants but fails to clearly report their quantitative architectural details (e.g., parameter counts). Adding a comparison table summarizing these details and computational characteristics (e.g., inference speed, memory usage) would improve transparency and help readers fairly interpret whether model size contributes to performance differences.

---

> ### Author Rebuttal · Authors · 2026-03-31
>
> We thank the reviewer for their thoughtful comments. The reviewer raises four weaknesses (W1, W2, W3, W4) and three questions (Q1, Q2, Q3).
>
> ---
> ## Responses to weaknesses
>
> **W1**: We do not wish to claim that current TSFMs are already capable of replacing hand-crafted features (HCFs) in scientific workflows. We will revisit and revise any language claiming that TSFMs are already fit to entirely replace HCFs. Our goal is instead to introduce standardized zero-shot benchmarking of TSFM on astronomical time series to give both the ML and Astro communities a clearer picture of what these models can do in/for astronomy and what technical gaps remain. One of our key results shows that TSFMs unilaterally outperform the astronomy-specific transformer models, providing guidance for improving model development in the astronomy literature.
>
> To your point, we also feel it is important to emphasize where TSFMs fall short to support TSFM development and continued performance gains in our benchmark. One major example of this is where we discuss the failure of TSFMs to properly treat irregular sampling. This warps important time series features (**Fig. 4**) and prevents the model from inferring the period (**Appx. H**), which we also showed to be the most important of the HCFs (**Appx. F**).
>
> ---
>
> **W2:** We have now added Chronos fine-tuning experiments in two settings:
> **(1) Chronos-FTfcst**, i.e., continued pretraining on our dataset with the original Chronos forecasting objective
> **(2) Chronos-FTcls**, i.e., end-to-end classification fine-tuning with all encoder layers unfrozen and an MLP classification head.
> The results are informative: **(1) does not improve downstream classification and is slightly worse than zero-shot Chronos**, whereas **(2) improves 3 of 4 MLP metrics over zero-shot Chronos** (Precision/Recall/F1). The end-to-end classification fine-tuned Chronos still remains below hand-crafted features, however. This suggests that adaptation can be helpful for Chronos but only when done directly. Forecasting fine-tuning fails due to the model not properly treating the irregular time steps. The model learns on the "wrong" information/ground truth. Classification fine-tuning is more promising. Even models pretrained on forecasting perform well through end-to-end classification finetuning. Figures illustrating full results are at this link: https://imgur.com/a/7kiMGXY.
>
> We also highlight that **Chronos-FTcls performs better than TSFMs designed for classification**. See response to **Reviewer kug1, W1** for the discussion.
>
> ---
>
> **W3:** We prioritize label credibility over broader, noisier class coverage. Many literature labels rely on low-capacity machine learning pipelines whose reliability varies dramatically and is not always well constrained. Instead, StarEmbed is built on the expert-generated CSPVS catalog to ensure a gold-standard foundation. The seven classes that are present have applications to a variety of pressing astrophysical problems, so model performance on them is of special importance.
>
> We focus specifically on ZTF because it provides the optimal combination of a long temporal baseline (~8 years), high-precision calibration, and multi-band coverage. Some or all of these qualities are absent in many other observatory datasets, including the Catalina Sky Survey data which was used to create the CSPVS.
>
> Transients fundamentally differ from periodic variables in both physics and methodology. Transient classification requires real-time analysis of partial events, whereas variable star science relies on full periodic light curves. Consequently, domain-specific models are rarely designed to cross this boundary.
>
>
> ---
>
> **W4:** We added new tables providing all the information (https://imgur.com/a/ThoAbY4). **Table 21** is the summary, while Tables 17 and 18 provide further details on the FLOPs calculation. We report FLOPs instead of inference speed to avoid the impact of hardware and implementation differences.
>
> ---
> ## Responses to questions
>
> **Q1:** We initially conducted fine-tuning only on Moirai because it offered any-variate capabilities unlike Chronos. However, we agree with the reviewer that it is important and interesting to fine-tune Chronos given its superior performance. In **W2** we reported the results from two different fine-tuning experiments.
>
> ---
>
> **Q2:** Please refer to **response to W4**.
>
> ---
>
> **Q3**: We agree and have extended this analysis to include classification results across the full Chronos and Moirai model families. Larger models with larger embeddings show similar classification performance, and they still do not outperform handcrafted features. See https://imgur.com/a/XdCSxFv for the full results.
>
> Our initial exploration of the "small" models was to make a more direct comparison between TSFMs and astronomy-specific transformers (Astromer uses ~5.4 M parameters).
>
> ---
>
> Thanks again for your time and effort! We look forward to further feedback and discussions.

---

> > ### Author Rebuttal · Reviewer_knTX · 2026-04-04
> >
> > Thanks the authors for the explanation. Although additional experiments are provided, I quention why not consider linear probing, partial and PEFT-style tuning as raised in my original comments? End-to-end finetuing with all layers unfrozen is usually not a good choice for adapting TSFMs. Therefore the conclusions derived from current experiments are not convicing enough.
> >
> > A major revision with more experiments may be needed to fully resolve this problem, so I keep my score in current state.

---

> > > ### Author Response · Authors · 2026-04-07
> > >
> > > We appreciate your acknowledgement of our new fine-tuning experiments and understand your follow-up requests. Linear probing is certainly an important setting to test the information content of the models' embeddings. As such, **we included linear probing in our original classification task results (line 305 - 306, Table 3)**, but we did not extend it to our fine-tuning experiments. Although what the best technique for fine-tuning a TSFM still remains an open question, partial and PEFT-style tuning are indeed very relevant tests. Below, we report on additional fine-tuning experiments on Chronos which now enable a comparison between **full fine-tuning** (following [1,2]), [**LoRA**](https://github.com/microsoft/LoRA) (the most common choice for PEFT), and **partial fine-tuning** (only tuning the LayerNorms, following [3]). We run each of these experiments both with an **MLP head** and a **linear head** to most comprehensively explore these techniques.
> > >
> > >
> > > ---
> > >
> > > **Linear Head + PEFT (LORA) / Partial Finetuning.**
> > >
> > > | Mode | Trainable Params | Macro F1 | Accuracy | Macro Prec | Macro Rec |
> > > |-|-|-|-|-|-|
> > > | Full FT | 8.4M (100%) | **0.661 ± 0.010** | **0.777 ± 0.010** | **0.615 ± 0.012** | **0.766 ± 0.005** |
> > > | LoRA | 102K (1.2%) | 0.622 ± 0.022 | 0.746 ± 0.023 | 0.572 ± 0.026 | 0.760 ± 0.006 |
> > > | LayerNorm | 9.2K (0.1%) | 0.622 ± 0.018 | 0.750 ± 0.029 | 0.571 ± 0.022 | 0.758 ± 0.004 |
> > >
> > > ---
> > >
> > > **MLP Head + PEFT (LORA) / Partial Finetuning.**
> > >
> > > | Mode | Trainable Params | Macro F1 | Accuracy | Macro Prec | Macro Rec |
> > > |-|-|-|-|-|-|
> > > | Full FT* | 9.6M (100%) | 0.659 ± 0.026 | **0.778 ± 0.028** | **0.613 ± 0.034** | 0.766 ± 0.005 |
> > > | LoRA | 1.3M (13.2%) | **0.661 ± 0.015** | 0.772 ± 0.020 | **0.613 ± 0.018** | **0.767 ± 0.006** |
> > > | LayerNorm | 1.2M (12.4%) | 0.648 ± 0.016 | 0.774 ± 0.023 | 0.605 ± 0.023 | 0.766 ± 0.006 |
> > >
> > >
> > > *Indicates the experiments done in the first round rebuttal.
> > > We conduct hyperparameter tuning and select the configuration with the highest macro-F1. Each metric is averaged over 10 random seeds runs. See https://imgur.com/a/xN1u5xr for the details.
> > >
> > >
> > > ---
> > >
> > > **We summarize the results as follows:**
> > >
> > > * The linear head results clearly reveal that full fine-tuning yields the best performance. With the MLP head, both full fine-tuning and LoRA yield comparable top performance, despite LoRA tuning only fewer than 1/7th as many parameters.
> > >
> > > * When only a small number of parameters are updated during fine-tuning, using a more expressive classification head becomes more important. In particular, under a linear head, both LoRA and LayerNorm tuning appear to be similarly constrained by the limited expressiveness of the head, resulting in lower F1 scores. **Under an MLP head, this bottleneck is alleviated by the nonlinear head. LoRA shows a clearer advantage over LayerNorm tuning (0.661 vs. 0.648 F1).** This is possibly because LoRA adapts the attention projections and thus better focuses on discriminative patterns in the time series, whereas LayerNorm-only tuning is limited to scale-and-shift updates.
> > >
> > > Thus, we think that PEFT-style fine-tuning, paired with an MLP head, offers a favorable performance–efficiency tradeoff. At the same time, it is important to note that **all of these fine-tuning variants still underperform hand-crafted features**. This is not a negative result; the primary goal of our study is to introduce a benchmark with a robust time-series dataset that the TSFM community has not yet been exposed to.
> > >
> > >  ---
> > >
> > > We also add more fine-tuning experiments with a forecasting loss. This aims to verify that the conclusion "forecasting-FT is worse than classification FT"(**response to W2**) is still true under new fine-tuning settings. Results show that the downstream classification on embeddings from the fine-tuned models still underperforms the zero-shot baseline **(See results table and more details: https://imgur.com/a/BFOCO4Y).**
> > >
> > > These confirm that the issue lies in the forecasting objective itself which is unsuitable for a model that does not treat the irregular sampling of our time series properly (see Fig. 4). Further, these also suggest that the issue does not lie in the hyperparameter selection or the adaptation strategy.
> > >
> > > ---
> > >
> > > After your comments, we believe that our study now convincingly demonstrates to the reader that current TSFMs (either in zero-shot or many fine-tuning settings) are not yet suitable to entirely replace old and conceptually simple techniques on this complex yet abundant time series data. The benchmark that we introduce, the first on astronomical time series, is a critical step toward alleviating this.
> > >
> > > ---
> > >
> > > [1] "Time-moe: Billion-scale time series foundation models with mixture of experts." ICLR (2025)
> > >
> > > [2] "This time is different: An observability perspective on time series foundation models." NeurIPS (2025)
> > >
> > > [3] "Visionts: Visual masked autoencoders are free-lunch zero-shot time series forecasters." ICML (2025)

---

### Decision · Program_Chairs · 2026-04-30

**Decision:**

Accept (regular)

**Comment:**

I recommend to accept the paper.

The paper introduces a new benchmark for time series foundation models, which is then used to benchmark and compare several families of foundation models across different tasks.

The contribution was found very worthwhile by all reviewers (and all but one clearly recommend to accept), with some concerns remaining about the benchmarking setup. One reviewer asked for experiments on "linear probing, partial and PEFT-style tuning" vs. end-to-end finetuning with some layers unfrozen which was reported by the authors in most of the cases. This issue can be addressed by incorporating the missing evaluation protocols and pointing readers explicitly to the fact that end-to-end finetuning was used in all but some cases; this should be added to at least partially address that reviewer's concern. New results run during the rebuttal phase should also be added to the paper (e.g. [the additional comparisons](https://openreview.net/forum?id=Ujf6Is4cdt&noteId=mwiY5uVyQ3)).

Other promises made to the reviewers in their rebuttals should likewise be checked and incorporated in the manuscript, including revising statements around applicability of TSFMs ("revisit and revise any language claiming that TSFMs are already fit to entirely replace HCFs").